# The Best of Both Worlds in Network Population Games: Reaching Consensus & Convergence to Equilibrium

**Shuyue Hu**
Shanghai Artificial Intelligence Laboratory
hushuyue@pjlab.org.cn

**Harold Soh**
National University of Singapore
harold@comp.nus.edu.sg

**Georgios Piliouras**
Singapore University of Technology and Design
georgios@sutd.edu.sg

## Abstract

Reaching consensus and convergence to equilibrium are two major challenges of multi-agent systems. Although each has attracted significant attention, relatively few studies address both challenges at the same time. This paper examines the connection between the notions of consensus and equilibrium in a multi-agent system where multiple interacting sub-populations coexist. We argue that consensus can be seen as an intricate component of intra-population stability, whereas equilibrium can be seen as encoding inter-population stability. We show that smooth fictitious play, a well-known learning model in game theory, can achieve both consensus and convergence to equilibrium in diverse multi-agent settings. Moreover, we show that the consensus formation process plays a crucial role in the seminal thorny problem of equilibrium selection in multi-agent learning.

## 1 Introduction

Two fundamental problems of multi-agent systems are *consensus formation* [26, 3, 12, 30, 5, 28, 86, 68] and *learning in games* [48, 33, 66, 58, 57, 70, 71, 13, 55]. The former (sometimes also referred to as cooperative control, flocking, synchronization, and social norms) aims to understand how a decentralized system of agents, which starts off in an initially disordered state, is able to develop shared coordinated beliefs, opinions, or behaviors [7]. The latter (also known as multi-agent learning) examines whether, how and what type of equilibrium arises as the result of the long-run learning processes of individual agents [33]. Each problem has attracted significant interest, yet, these two problems are largely studied in isolation.

Is there a unified way to think about consensus formation and learning in games? We argue that *consensus* and *equilibrium*, the fundamental notions of these two fields, can be both understood as *stability* concepts in a multi-agent system where there co-exist multiple interacting sub-populations (or populations hereafter). Consensus can be seen as an intricate component of *intra*-population stability that reflects whether each population is internally cohesive, i.e., a population is stable if all the agents belonging to this population achieve consensus and develop the same belief, opinion, or behavior. In contrast, game-theoretic (e.g., Nash) equilibrium can be seen at a more macro level as encoding *inter*-population stability, i.e., the stability between different populations is achieved by alignment to their population-wise incentives. Namely, a stable system of populations is characterized by the internal cohesion of each population as well as the game-theoretic equilibration between different populations. The merits of this way of thinking are twofold. It presents a non-trivial connection

37th Conference on Neural Information Processing Systems (NeurIPS 2023).

between consensus and equilibrium, bringing them together under the same umbrella of stability concepts for a system of populations. Moreover, it distinguishes between intra-population stability and inter-population stability so that the emerging phenomena that it can describe are significantly richer, and that it allows for important distinctions that would have otherwise been impossible.

In the rich literature on consensus formation, the idea of building up a connection to game-theoretic equilibration is not entirely new. Classic mathematical models, such as the Degroot model [27] and the Friedkin-Johnsen model [31], showed that consensus of opinions can be reached if individuals form their opinions taking a weighted average of their own opinions and the opinions of others. In seminal work, Bindel et al. [15] showed that such a repeated averaging process is equivalent to a best-response play in the game with a quadratic reward function measuring the distance between individuals' opinions, leading to interest in understanding consensus from the perspective of learning in games [38, 23, 2, 14, 36, 10, 9]. In parallel, Marden et al. [62] demonstrated how the cooperative control problem of consensus can be formulated as better reply with inertia dynamics in potential games, while Young [89] proved the emergence of social norms using evolutionary game theory. These notable advances, however, typically assumed the reward structure to encode an incentive to align individual agents' opinions or behaviors with those of others. Little is formally known about consensus formation in the absence of such a coordinative reward structure, or whether such a coordinative reward structure is a prerequisite for achieving both consensus formation and game-theoretic equilibration.

In the literature on multi-agent learning, the concept of consensus has been less explored compared to the abundant study of convergent or non-convergent learning behaviors in games [13, 79, 66, 81, 77, 6, 57, 16]. We argue that consensus is of crucial importance to multi-agent learning—it ensures not only intra-population stability when there co-exist multiple interacting populations but also complete predictability and computational efficiency in more general settings. Imagine the case that for a system of agents, the average probability ($\bar{x}_A$ and $\bar{x}_B$) of choosing between two strategies ($A$ and $B$) is shown to equilibrate at $\bar{x}_A = \bar{x}_B = 0.5$. If consensus is reached, it is completely predictable that every agent plays $A$ and $B$ with equal probability $0.5$. However, if consensus is not reached, there are many possibilities for agents' behaviors, to name a few—it is possible (i) that exactly half of the agents play $A$ with probability $1$ and the other half plays $B$ with probability $1$, and (ii) that some agents play $A$ and $B$ with equal probability $0.5$ whereas the rest constantly switches between these two strategies. The unpredictability in agents' behaviors can be problematic for system design, especially in safety-critical scenarios, where a system designer ought to rule out the possibility of getting the system to evolve into some (e.g., unsafe) states [74, 76, 73, 91]. From the perspective of computational efficiency, consensus is usually a desirable property and is embedded in the mean field game literature to simplify the characterization of equilibrium—by exploiting the homogeneity of the population, it suffices to understand the behavior of a single representative agent facing the mean field [55, 54, 51, 71, 42, 87, 1].

Putting these together motivates our central questions:

- *Are there natural multi-agent learning models that can achieve the best of both worlds— reaching consensus as well as convergence to equilibrium—in diverse settings?*

- *How does the consensus formation process affect equilibrium selection in multi-agent learning?*

**Our Model.** We provide an affirmative answer to the first question, and our solution is—*smooth fictitious play* (SFP) [32]—a well-known belief-based learning model in game theory. In this paper, we consider a *network of populations*, where each vertex represents a population (or continuum) of agents and each edge denotes that agents from connected populations are paired up to play 2-player subgames. We call this type of game a network population game,[1] as it has a close connection to the network (or graphical polymatrix) games [52] and the population games [75]. By SFP, each agent forms beliefs about the strategies of agents of the connected populations based on the average

---

[1]A good survey for network population games and variants thereof is [83]. A prototypical example is the men-women interactions that give rise to a 2-population Battle of the Sexes game, which corresponds to a single-edge network in our case. Similarly, one can think of an ecosystem with a graph of predator-prey interactions; the fact that there is no self-interaction within a population captures that there is no in-species cannibalism. Another example is the multi-army combat setting; combatants of the same army do not face each other.

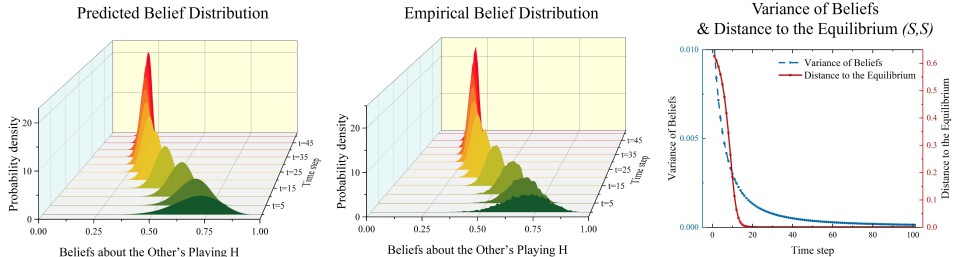

(a) The initial beliefs are distributed according to Beta$(14, 6)$ for both populations. The variance of initial beliefs is $0.01$ and the mean of initial beliefs is $0.7$.

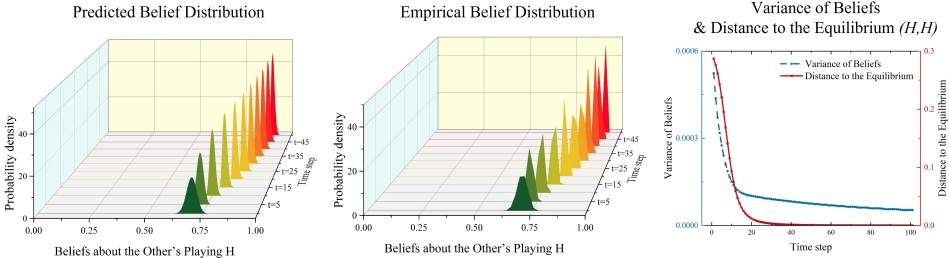

(b) The initial beliefs are distributed according to Beta$(280, 120)$ for both populations. The variance of initial beliefs is $\approx 0.0005$ and the mean of initial beliefs is $0.7$.

Figure 1: System evolution in 2-population stag hunt games. Over time, the belief distributions become more concentrated (left panel: the predictions of our continuity equations, middle panel: the empirical distributions in agent-based simulations), and the variance of beliefs and the distance (L1 norm) to the equilibrium are both decreasing (right panel). Even though the consensus formation process starts with the same initial mean belief, a small variance of initial beliefs results in the risk dominant equilibrium $(H, H)$ (subplot (b)), whereas a large variance of initial beliefs results in the payoff dominant equilibrium $(S, S)$ (subplot (a)). A video for the evolution of these two belief distributions is available at *https://sites.google.com/view/shuyue-hu*.

plays of those populations, and responds with an aggregated smoothed best response. To allow for a meaningful study of consensus formation, we assume that agents within each population have different beliefs and different behaviors in the initial state. In other words, our model is a *heterogeneous belief* model [35], unlike the homogeneous belief models extensively studied in the SFP literature [8, 82, 11, 32, 44, 47, 46].

**Our Approach & Key Results.** To see whether a consensus is reached, one needs a measure of it. A natural approach is to assume the system state to be a probability measure (or distribution) over the set of beliefs (referred to as the belief distribution). This notion of a system state allows us to quantify how much agents reach a consensus by examining the *variance* of the belief distribution. Intuitively, a value of zero for the variance means that agents have the same belief and have achieved consensus. However, analyzing the evolution of the belief distribution requires new techniques beyond the classic approach of stochastic approximation [8, 11, 35, 44, 47]. We address this challenge by establishing a system of nonlinear partial differential equations—*continuity equations*[2]—to describe the continuous-time change in the belief distribution (Proposition 1). This approach allows us to analyze the evolution of the variance without deriving an analytical form of the belief distribution. We show that for *any* network population games, the variance of beliefs decreases to zero at a quadratic rate with time in every population (Theorem 1). This means that even if the 2-player subgames that agents (of different populations) play are not coordinative but zero-sum, consensus will still be reached. Put differently, a coordinative reward structure is not a prerequisite for achieving consensus.

---

[2]A continuity equation is a PDE commonly encountered in physics, which describes the transport phenomena of some quantity (e.g., mass, energy, momentum and other conserved quantities) in a physical system.

The next step is to examine the convergence to equilibrium. Our finding that the variance of beliefs decreases to zero suggests that the asymptotic behavior of a population that starts with different beliefs can be understood by studying that of a single representative agent. Inspired by this, we temporarily override the assumption of the heterogeneous belief model and assume a single representative agent for each population, resulting in the classic form of network games with each vertex representing a single agent. Under this assumption, we use Lyapunov arguments to prove the convergence of SFP to quantal response equilibria (QRE)[3] in two classes of network games that capture network competition and network coordination. Specifically, we show that for a weighted zero-sum network game, SFP converges to a unique QRE even if the underlying game has many distinct Nash equilibria (Theorem 2). We also show that for an exact potential network game that exhibits a star structure, SFP converges to the set of QRE (Theorem 3). We then use the techniques of asymptotically autonomous dynamical systems to show that the convergence to QRE holds for network population games as well (Theorem 4 and Theorem 5). Therefore, SFP converges to QRE in all weighted zero-sum network (population) games as well as all exact potential star-graph (population) games.

At this point, we have shown that SFP can achieve consensus as well as convergence to equilibrium in a wide range of network population games. Specifically, consensus is always reached, regardless of the specific setting. In the language of stability concepts, by SFP, the intra-population stability is always guaranteed, but the inter-population stability may require additional assumptions about the incentive nature as well as the network topology. This may lead to an error-prone intuition that the consensus formation process is orthogonal to the equilibrium selection process and thus has little effect on it. This is indeed *not* true. We illustrate this through an example of a 2-population stag hunt game, where different variances of initial beliefs result in qualitatively different long-term system behaviors. As we show in Figure 1, even though the initial mean belief is the same, the consensus formation process starting with a low level of initial disagreement (a small value for initial variance) results in the risk dominant equilibrium $(H, H)$, while starting with a high level of initial disagreement (a large value for initial variance) leads to the payoff dominant equilibrium $(S, S)$. We provide additional evidence covering all possible initial mean beliefs under the same 2-population stag-hunt game setting in Figure 2 (Section 6). Figure 2 shows that increasing the variance of initial beliefs from 0 to 0.02, 0.05, and 0.1 expands the region of attraction of the payoff dominant equilibrium $(S, S)$, allowing a wider range of initial mean beliefs to approach it. Thus, in the case of network coordination, consensus formation plays a crucial role in the seminal thorny problem of equilibrium selection, and a larger initial variance can promote convergence to the equilibrium $(S, S)$. In the case of network competition, convergence to a unique QRE is formally guaranteed (Theorem 4); we empirically verify this in 5-population asymmetric matching pennies games with many distinct Nash equilibria (Section 1 of the Appendix).

**Other Related Work.** SFP is a well-studied model that converges in various classes of games, including most 2-player-2-action games [32, 44, 11, 47], n-player potential game s[82, 45], and zero-sum and identical-interest stochastic games [8, 78, 59]. However, SFP in network (population) games has not been previously explored. Our research extends the findings of Fudenberg and Takahashi [35] (who showed that agents with different beliefs in a 2-population setting will converge to the same belief) to the network settings, but we demonstrate that the variance of initial beliefs plays a crucial role in equilibrium selection, which has not been found in [35]. On the other hand, networked multi-agent learning is a current frontier in AI research [90, 61, 43, 21, 17, 4, 25]. Previous studies have shown the non-convergent behavior of replicator dynamics in (weighted) zero-sum network games [72, 18, 66, 79], the convergence of smooth Q-learning in weighted potential [56] and weighted zero-sum network games [57], and the convergence of fictitious play in zero-sum network games [29]. This paper provides new convergence results for SFP in both network games and network population games, covering a wide range of generic 2-population-2-action settings, star-graph settings that admit an exact potential, and all weighted zero-sum settings. Finally, the evolution of probability distributions over initial conditions or strategies has also received increasing interest [22, 53, 49, 50, 24, 60]. Continuity equations have been used as a tool to show that under Cross learning, agents with different mixed strategies in a population game will not converge to the same mixed strategy [53], and to also describe the more complicated Q-learning dynamics in various settings [49, 50, 24, 60]. This paper formally shows that the probability distribution over

---

[3]Quantual response equilibrium is a game theoretic solution concept under bounded rationality [65].

initial conditions can eventually degenerate to a point mass, and leveraging on this, presents a novel technique for proving the convergence of learning dynamics.[4]

## 2  Preliminaries

A **network population game** $\Gamma = (N, (V, E), (S_i, \omega_i)_{\forall i \in V}, (\mathbf{A}_{ij})_{(i,j) \in E})$ consists of a multi-agent system $N$ distributed over a connected graph $(V, E)$, where $V = \{1, ..., n\}$ is the set of vertices each represents a population (continuum) of agents, and $E$ is the set of pairs, $(i, j)$, of population $i \neq j \in V$. For each population $i \in V$, agents of this population has a finite set $S_i$ of pure strategies (or actions) with generic elements $s_i \in S_i$. Agents may also use mixed strategies (or choice distributions). For an arbitrary agent $k$ in population $i$, its mixed strategy is a vector $\mathbf{x}_i(k) \in \Delta_i$, where $\Delta_i$ is the simplex in $\mathbb{R}^{|S_i|}$ such that $\sum_{s_i \in S_i} x_{is_i}(k) = 1$ and $x_{is_i}(k) \geq 0, \forall s_i \in S_i$. Each edge $(i, j) \in E$ defines a series of two-player subgames between populations $i$ and $j$. We denote the payoff matrices for agents of population $i$ and $j$ in these two-player subgames by $\mathbf{A}_{ij} \in \mathbb{R}^{|S_i| \times |S_j|}$ and $\mathbf{A}_{ji} \in \mathbb{R}^{|S_j| \times |S_i|}$, respectively. For every time step, each agent chooses a (mixed or pure) strategy and plays that strategy in all two-player subgames with agents from the connected (or neighbor) populations. Let $\mathbf{x} = (\mathbf{x}_i, \{\mathbf{x}_j\}_{(i,j) \in E})$ be a mixed strategy profile, where $\mathbf{x}_i$ (or $\mathbf{x}_j$) denotes a generic mixed strategy in population $i$ (or $j$). Given the mixed strategy profile $\mathbf{x}$, the expected payoff of using $\mathbf{x}_i$ in the game $\Gamma$ is

$$r_i(\mathbf{x}) = r_i(\mathbf{x}_i, \{\mathbf{x}_j\}_{(i,j) \in E}) := \sum_{(i,j) \in E} \mathbf{x}_i^\top \mathbf{A}_{ij} \mathbf{x}_j. \tag{1}$$

**Smooth (or stochastic) fictitious play** (SFP) is a belief-based model for learning in games. By SFP, agents act as if they are Bayesian such that they assume that the strategies of other agents are drawn from some fixed but unknown distribution. Given a game $\Gamma$, consider an arbitrary agent $k$ in a population $i \in V$. Let $V_i = \{j \in V : (i, j) \in E\}$ be the set of neighbor populations. Agent $k$ maintains a weight $\kappa^i_{js_j}(k)$ for each opponent strategy $s_j \in S_j$ of each neighbor population $j \in V_i$. For every time step $t$, agents refine these weights based on the average play of each neighbor population, i.e., for each opponent strategy $s_j \in S_j, j \in V_i$,

$$\kappa^i_{js_j}(k, t+1) = \kappa^i_{js_j}(k, t) + \bar{x}_{js_j}(t), \tag{2}$$

where $\bar{x}_{js_j}$ is the mean probability of playing strategy $s_j$ in population $j$.[5] For simplicity, we assume the initial sum of weights $\sum_{s_j \in S_j} \kappa^i_{js_j}(k, 0)$ to be the same for every agent in the system and denote this initial sum by $\lambda$. Based on the weights, agent $k$ forms a belief about the neighbor population $j$ such that each opponent strategy $s_j$ is played with probability

$$\mu^i_{js_j}(k) = \frac{\kappa^i_{js_j}(k)}{\sum_{s'_j \in S_j} \kappa^i_{js'_j}(k)}. \tag{3}$$

Let $\boldsymbol{\mu}^i_j(k)$ be the vector of beliefs with the $s_j$-th element equals $\mu^i_{js_j}(k)$. Given a game $\Gamma$, agent $k$'s expected payoff of using a pure strategy $s_i \in S_i$ in response to the aggregate beliefs $\boldsymbol{\mu}^i_j(k), \forall j \in V_i$ about neighbor populations is

$$u_{is_i}(k) = \sum_{j \in V_i} \mathbf{e}_{s_i}^\top \mathbf{A}_{ij} \boldsymbol{\mu}^i_j(k), \tag{4}$$

where $\mathbf{e}_{s_i}$ is a unit vector with the $s_i$-th element equal 1. By convention [33, 34, 44], we suppose that agents' payoffs are perturbed and that agents play smooth best responses maximizing their perturbed

---

[4]Our approach highlights the consensus property as a valuable tool for establishing the convergence of learning under population settings. For future research that studies learning dynamics under population settings, one can find inspiration in our approach by initially verifying the consensus property and subsequently leveraging this property to establish convergence results.

[5]Observing the average play of other populations is natural in settings with a large number of agents and can be viewed as *mean-field* observations [88, 37]; for example, the average household income of each state is made publicly available, and people form beliefs about the behaviors of taxi drivers vs non-professional drivers after observing the numerous driving behaviors on the road.

payoffs. Agent $k$'s perturbed payoff of using a mixed strategy $\mathbf{x}_i(k) \in \Delta_i$ is defined as

$$\pi_i\left(\mathbf{x}_i(k), \{\boldsymbol{\mu}_j^i(k)\}_{j \in V_i}\right) = \sum_{j \in V_i} \mathbf{x}_i(k)^\top \mathbf{A}_{ij} \boldsymbol{\mu}_j^i(k) + \epsilon v\left(\mathbf{x}_i(k)\right), \tag{5}$$

where $\epsilon > 0$, and the function $v$ is strictly concave and its gradient becomes arbitrarily large near the boundary of the simplex $\Delta_i$ [44].[6] For the function $v$, a typical choice is $v\left(\mathbf{x}_i(k)\right) = -\frac{1}{\beta} \sum_{s_i \in S_i} x_{is_i}(k) \ln(x_{is_i}(k))$. Under this form, the maximization of the perturbed payoff $\pi_i$ yields a unique choice distribution $\mathbf{x}_i(k)$ from the logit choice rule [45], that is,

$$x_{is_i}(k) = \frac{\exp(\beta u_{is_i}(k))}{\sum_{s_i' \in S_i} \exp(\beta u_{is_i'}(k))}, \tag{6}$$

where $\beta$ is the temperature (or the degree of rationality). For clarity of our presentation, we choose to focus on the logit choice rule in this paper; however, all our results readily generalize to any function $v$ satisfying the above two standard assumptions. In the following, we usually drop the time index $t$ and agent index $k$ in the bracket (depending on the context) for notational convenience.

## 3  Belief Dynamics in Network Population Games

Observe that for an arbitrary agent $k$ of population $i$, its belief $\boldsymbol{\mu}_j^i(k)$ about population $j$ is in the simplex $\Delta_j = \{\boldsymbol{\mu}_j^i(k) \in \mathbb{R}^{|S_j|} | \sum_{s_j \in S_j} \mu_{js_j}^i(k) = 1, \mu_{js_j}^i(k) \geq 0, \forall s_j \in S_j\}$. We assume that the system state is characterized by a Borel probability measure $P$ defined on the state space $\Delta = \prod_{i \in V} \Delta_i$. Given $\boldsymbol{\mu}_i \in \Delta_i$, we write the marginal probability density function as $p(\boldsymbol{\mu}_i, t)$. Note that $p(\boldsymbol{\mu}_i, t)$ represents the density of agents having the belief $\boldsymbol{\mu}_i$ *about* population $i$ *throughout the system*; it does *not* represent the density of agents of population $i$ that have the belief $\boldsymbol{\mu}_i$, *nor* the density of agents having the belief $\boldsymbol{\mu}_i$ about population $i$ in a certain population. Define $\boldsymbol{\mu} = \{\boldsymbol{\mu}_i\}_{i \in V} \in \Delta$. Since agents maintain separate beliefs about different neighbor populations, the joint probability density function $p(\boldsymbol{\mu}, t)$ can be factorized, i.e., $p(\boldsymbol{\mu}, t) = \prod_{i \in V} p(\boldsymbol{\mu}_i, t)$. We make the following assumption for the initial marginal density functions.

**Assumption 1.** *At time $t = 0$, for each population $i \in V$, the marginal density function $p(\boldsymbol{\mu}_i, t)$ is continuously differentiable and has zero mass at the boundary of the simplex $\Delta_i$.*

This assumption excludes the setting that a population of agents starts off with the same belief as well as the setting that agents start off with extreme beliefs. Under this mild condition, we derive a partial differential equation to describe the evolution of the belief distribution.

**Proposition 1.** *The continuous-time dynamic of the marginal density function $p(\boldsymbol{\mu}_i, t)$ for each population $i \in V$ is described by a partial differential equation*

$$-\frac{\partial p(\boldsymbol{\mu}_i, t)}{\partial t} = \nabla \cdot \left( p(\boldsymbol{\mu}_i, t) \frac{\bar{\mathbf{x}}_i - \boldsymbol{\mu}_i}{\lambda + t + 1} \right) \tag{7}$$

*where $\nabla \cdot$ is the divergence operator and $\bar{\mathbf{x}}_i$ is the mean mixed strategy with each $s_i$-th element*

$$\bar{x}_{is_i} = \int_{\prod_{j \in V_i} \Delta_j} \frac{\exp\left(\beta u_{is_i}\right)}{\sum_{s_i' \in S_i} \exp\left(\beta u_{is_i'}\right)} \prod_{j \in V_i} p(\boldsymbol{\mu}_j, t) \left( \prod_{j \in V_i} d\boldsymbol{\mu}_j \right) \tag{8}$$

*where $u_{is_i} = \sum_{j \in V_i} \mathbf{e}_{s_i}^\top \mathbf{A}_{ij} \boldsymbol{\mu}_j$.*

The partial differential equation in Equation 7 is also known as a *continuity equation* in the study of transport phenomena (e.g., of mass or energy) in a physical system. We remark that continuity equations, tracking the evolution of a probability distribution, do not allow for general solutions. However, continuity equations provide a tractable way to examine the evolution of the moments of the probability distribution, without obtaining an analytical form of that distribution. The analytical form for the dynamic of the variance of beliefs (the second moment of the belief distribution) is given as follows.

---

[6]Alternatively, SFP can be defined through a truly stochastic perturbation of payoffs [32].

**Theorem 1.** *The dynamic of the variance of beliefs $\boldsymbol{\mu}_i$ about each population $i \in V$ is governed by an ordinary differential equation such that for each strategy $s_i$,*

$$\frac{d\text{Var}(\mu_{is_i})}{dt} = -\frac{2\text{Var}(\mu_{is_i})}{\lambda + t + 1}. \tag{9}$$

*At given time t, $\text{Var}(\mu_{is_i}) = \left(\frac{\lambda+1}{\lambda+t+1}\right)^2 \sigma^2(\mu_{is_i})$, where $\sigma^2(\mu_{is_i})$ is the initial variance. Thus, the variance $\text{Var}(\mu_{is_i})$ decays to zero at a quadratic rate with time.*

Note that the above theorem makes no assumption about the 2-player subgames agents play, the network structure of populations, or even the initial belief distribution. Therefore, it shows that for any network of populations, the variance of beliefs about each population always asymptotically tends to zero. As an immediate consequence of this theorem, the asymptotic distribution of the beliefs is as follows:

**Corollary 1.** *The probability density function $p(\boldsymbol{\mu}_i, t)$ for every population $i \in V$ will eventually evolve into a Dirac delta function, implying that in the long run agents reach a consensus and play the same mixed strategy within each population.*

The intuition of Theorem 1 and Crollary 1 is that under SFP, agents revise their beliefs about other populations based on the average play of those populations; consequently, the observations are the same within an agent population, and thus it is only a matter of time that the agent population will develop the same belief and play the same strategy. Once consensus is reached, agents within a population will not develop different beliefs afterwards, as ensured by Theorem 1. In this sense, the agent population is internally cohesive and achieves intra-population stability.

However, it does not mean that different agent populations will eventually play the same strategy. This is because agents' strategies are the smooth best responses to their aggregated beliefs about their own set of neighbor populations (which generally vary depending on the network topology). This naturally raises the question of how different agent populations align with their population-wise incentives (which can be of a competitive nature). This also sets our paper apart from previous work that studies game-theoretic equilibration in the context of consensus formation [15, 62, 38, 23, 2, 14, 36, 10, 9], in which consensus is achieved as the result of agents' incentives to align their opinions or behaviors to those of others (i.e., the incentives are of a coordinative nature).

Before diving into the convergence results, we take a moment to analyze the mean belief (first moment) dynamic. Intuitively, the mean belief represents the average system state. The following proposition provides an analytical form of the mean belief dynamic.

**Proposition 2.** *The mean belief dynamic for each population $i \in V$ is governed by an ordinary differential equation such that for each strategy $s_i$,*

$$\frac{d\bar{\mu}_{is_i}}{dt} \approx \frac{f_{s_i}(\{\boldsymbol{\mu}_j\}_{j \in V_i}) - \bar{\mu}_{is_i}}{\lambda + t + 1} + \frac{\sum_{j \in V_i} \sum_{s_j \in S_j} \frac{\partial^2 f_{s_i}(\{\boldsymbol{\mu}_j\}_{j \in V_i})}{(\partial \mu_{js_j})^2} \text{Var}(\mu_{js_j})}{2(\lambda + t + 1)}, \tag{10}$$

*where $f_{s_i}(\{\boldsymbol{\mu}\}_{j \in V_i})$ is the logit choice rule (Equation 6) applied to strategy $s_i \in S_i$, and $\text{Var}(\mu_{js_j})$ is the variance of belief $\mu_{js_j}$ in the entire system.*

We remark that the mean belief dynamic, to be exact, should be under the joint influence of infinitely many moments of the belief distribution. However, the higher the order of the moments, the less of the influence of the moments, and by convention, the effects of the third and higher moments can be assumed negligible [39, 41, 64].[7] Thus, in this proposition, the mean belief dynamic depends on the current mean belief and also on the current variance of beliefs.

Recall that Theorem 1 shows that the variance of beliefs will eventually decrease to zero. As a result, the second term of the mean belief dynamic (Equation 10) will asymptotically tend to zero. This suggests that the asymptotic mean belief dynamic of a population that starts with different beliefs in the initial state might be understood by studying the belief dynamic of a single representative agent.

---

[7]This is formally known as the moment closure approximation [39, 41, 64], which is a common method used to estimate moments of population models. As we detail in the appendix, the effects of the higher moments are counter-balanced by their coefficients as a result of Taylor expansion; moreover, the beliefs $\mu_{is_i}, \forall i \in V, \forall s_i \in S_i$ are between 0 and 1, which also lead to small values of higher moments.

Intuitively, if each population can be represented by a single agent, then the concerned network population games will effectively degenerate into the classic form of network games with each vertex representing a single agent. Inspired by this, in the next section, we temporarily focus on SFP in the classic network games.

## 4 Convergence of Smooth Fictitious Play in Classic Network Games

In a classic network game (or network game for short), the only difference with respect to the network population games is that now each vertex represents a single agent. We derive the belief dynamic of each vertex (or agent) in the following proposition.

**Proposition 3.** *In a network game, the belief dynamic of each agent is governed by an ordinary differential equation such that for each agent $i$ and each strategy $s_i$,*

$$\frac{d\mu_{is_i}}{dt} = \frac{f_{s_i}(\{\boldsymbol{\mu}_j\}_{j \in V_i}) - \mu_{is_i}}{\lambda + t + 1} = \frac{x_{is_i} - \mu_{is_i}}{\lambda + t + 1}. \tag{11}$$

It follows from this proposition that the equilibrium state should satisfy the following property.

**Proposition 4.** *In a network game, the equilibrium state should satisfy that $\mathbf{x}_i^* = \boldsymbol{\mu}_i^*$ for every strategy $s_i \in S_i$ and every population $i \in V$. Thus, the equilibrium state corresponds to the solution to the system of equations*

$$x_{is_i}^* = \frac{\exp\left(\beta \sum_{j \in V_i} \mathbf{e}_{s_i}^\top \mathbf{A}_{ij} \mathbf{x}_j^*\right)}{\sum_{s_i' \in S_i} \exp\left(\beta \sum_{j \in V_i} \mathbf{e}_{s_i'}^\top \mathbf{A}_{ij} \mathbf{x}_j^*\right)}, \quad \forall s_i \in S_i, \forall i \in V. \tag{12}$$

*Such an equilibrium state coincides with the Quantal Response Equilibria [65] of the network game.*

By QRE, in this paper, we refer to their canonical form also referred to as logit QRE in the literature [40]. We remark that QRE [65] is a game theoretic solution concept under bounded rationality, and is an extension of the standard Nash equilibrium which allows for errors in choice.

**Weighted zero-sum network games** capture the case of network competition, and are the generalization of 2-player zero-sum games to the network settings [19]. A network game is weighted zero-sum if there exist positive constants (or weights) $\omega_1, \ldots, \omega_n$ such that

$$\sum_{i \in V} \omega_i r_i(\mathbf{x}) = \sum_{(i,j) \in E} \left(\omega_i \mathbf{x}_i^\top \mathbf{A}_{ij} \mathbf{x}_j + \omega_j \mathbf{x}_j^\top \mathbf{A}_{ji} \mathbf{x}_i\right) = 0, \quad \forall \mathbf{x} \in \prod_{i \in V} \Delta_i, \tag{13}$$

where $r_i(\mathbf{x})$ is the expected payoff of using a mixed strategy $\mathbf{x}_i$ given the mixed strategy profile $\mathbf{x} = (\mathbf{x}_i, \{\mathbf{x}_j\}_{(i,j) \in E})$. If subgames along every edge $(i,j) \in E$ are 2-player zero-sum, i.e., $\mathbf{x}_i^\top \mathbf{A}_{ij} \mathbf{x}_j + \mathbf{x}_j^\top \mathbf{A}_{ji} \mathbf{x}_i = 0$, then the network game is zero-sum; however, the converse is not generally true. As opposed to 2-player zero-sum games that admit a unique value at equilibrium [85], the Nash equilibrium payoffs in weighted zero-sum network games need not be unique, and it generally allows for infinitely many Nash equilibria. In the following theorem, we show that regardless of the number of Nash equilibria in a network game, the belief dynamic is guaranteed to converge to a unique QRE.

**Theorem 2.** *For any weighted zero-sum network game, the belief dynamic converge to a unique QRE which is globally asymptotically stable.*

**Exact potential network games** capture the case of network coordination. A network game admits an exact potential if for each edge $(i,j) \in E$, the payoff matrices of the two-player subgame satisfy $\mathbf{A}_{ij} = \mathbf{A}_{ji}^\top$ [21, 20]. Pure or mixed Nash equilibria in network coordination are complex. As reported in recent works [20, 17, 4], finding a pure Nash equilibrium is PLS-complete. Hence, learning in the general case of network coordination is difficult and generally requires some conditions [67, 69]. In the following theorem, we show the convergence result in some cases of network coordination.

**Theorem 3.** *For any exact potential network game that exhibits a star structure, each orbit of the belief dynamic converges to the set of QRE.*

We remark that we employ the Lyapunov approach (i.e., establishing the existence of the Lyapunov function) to prove the above convergence results. As a result, the convergence to equilibrium is equivalent to the network of agents being stable in the sense of Lyapunov.

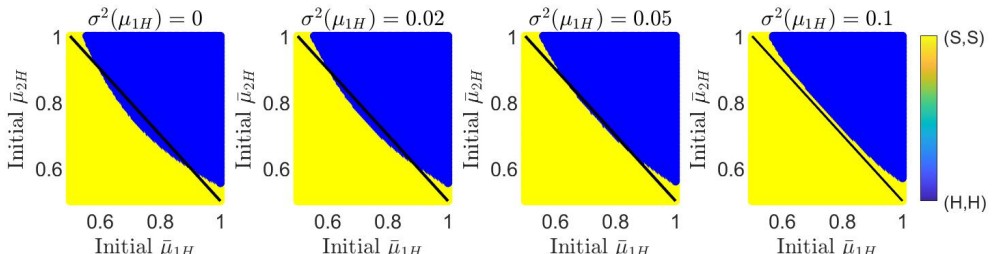

Figure 2: Initial disagreement helps select the payoff dominant equilibrium $(S, S)$ (yellow: the equilibrium $(S, S)$, blue: the equilibrium $(H, H)$). As the variance of initial beliefs (denoted by $\sigma^2(\mu_{1H})$) increases from the left to right panel, a wider range of initial mean beliefs will approximately reach the equilibrium $(S, S)$ in the limit. For simplicity, we set $\sigma^2(\mu_{1H}) = \sigma^2(\mu_{2H})$. The black line on the diagonal is there for ease of comparison across different panels.

## 5  Convergence of Smooth Fictitious Play in Network Population Games

We shall now formalize that the convergence results in network games can be seamlessly carried over to network population games with different beliefs in the initial state, and establish the inter-population stability in network population games. Recall that the mean belief dynamic (Proposition 2) depends on the current mean belief as well as the current variance of beliefs, with the variance decreasing over time (Theorem 1). Leveraging on the definition of *asymptotically autonomous* dynamical systems [63], we establish the following lemma.

**Lemma 1.** *In a network population game, for every population $i \in V$, the mean belief dynamic (Equation 10) is asymptotically autonomous with the limit equation being $\frac{d\mu_i}{dt} = \mathbf{x}_i - \mu_i$, which after time-reparmeterization is equivalent to the belief dynamic in a network game (Equation 11).*

Thieme [84] provided the following seminal result that formally connects the asymptotic behavior of an asymptotically autonomous dynamical system to that of its associated limit equation.

**Lemma 2** (Thieme [84] Theorem 4.2). *Given a metric space $(X, d)$, let the solution flows of an asymptotically autonomous system and its limit equation be $\phi$ and $\Theta$, respectively. Assume that the equilibria of $\Theta$ are isolated compact $\Theta$-invariant subsets of $X$. The $\omega$-$\Theta$-limit set of any pre-compact $\Theta$-orbit contains a $\Theta$-equilibrium. The point $(s, x), s \geq t_0, x \in X$, have a pre-compact $\phi$-orbit. Then the following alternative holds: 1) $\phi(t, s, x) \rightarrow e, t \rightarrow \infty$, for some $\Theta$-equilibrium $e$, and 2) the $\omega$-$\phi$-limit set of $(s, x)$ contains finitely many $\Theta$-equilibria which are chained to each other in a cyclic way.*

Combining the above lemmas and the convergence results under the setting of classic network games, the convergence results of SFP in network population games are readily deducible:

**Theorem 4.** *For any weighted zero-sum network population game, the belief (as well as the choice distribution) of every individual converges to a unique QRE.*

**Theorem 5.** *For any exact potential network population game that exhibits a star structure, the belief (as well as the choice distribution) of every individual converges to the set of QRE.*

Note that under the 2-population settings, there is only one possible graph structure—one edge connecting the two populations. Hofbauer and Hopkins [44] have shown that SFP converges in all 2-player weighted potential games, and observed that almost all 2-player-2-action games are either 2-player weighted potential games or 2-player weighted zero-sum games. Therefore, combining their results and our aforementioned results, SFP converges to the set of QRE under almost all the 2-population-2-action settings (including all the 2-population-2-action weighted potential settings and all the 2-population-2-action weighted zero-sum settings).

## 6  Experiments: Equilibrium Selection in Two-Population Stag Hunt Games

In a two-population stag hunt game, agents from two populations are paired up to choose between two actions $H$ (hare) and $S$ (stag). If both players hunt a hare, they both receive a reward of 1. If

both players hunt a stag, they both receive a reward of 4. If one hunts a hare and the other hunts a stag, the former receives a reward of 2 whereas the latter receives nothing. There are two pure strategy Nash equilibria in each subgame: the risk dominant equilibrium $(H, H)$ and the payoff dominant equilibrium $(S, S)$, which naturally raises the problem of equilibrium selection.

Figure 1 showed that given the same initial mean belief, changing the variances of initial beliefs can result in different limit behaviors. In Figure 2, we systematically studied the effect of the variance of initial beliefs by visualizing how it affects the regions of attraction to different equilibria. To be more specific, we numerically solved the mean belief dynamics for a wide range of initial mean beliefs, given different variances of initial beliefs, and colored the regions of attraction. We observed in Figure 2 that as the variance of initial beliefs increases (from the left to right panel), a wider range of initial mean beliefs results in the convergence to the QRE that approximates the payoff dominant equilibrium $(S, S)$. Hence, consensus is inevitable, but the initial disagreement still provides an approach to equilibrium selection and helps select the highly desirable equilibrium.

Note that in the case of network competition, the convergence to a unique QRE is formally guaranteed; we empirically verified this through a 5-population asymmetric matching pennies game presented in Appendix A.

## 7 Conclusions

This paper presents a formal treatment for SFP in network population games where each population of agents start off with different beliefs and strategies. Representing the system state with a distribution over beliefs, we show that consensus is always reached, and that agents eventually pointwise converge to the same belief, regardless of specific settings. In addition, we establish the convergence of SFP to Quantal Response Equilibria in general competitive network population games as well as exact potential network population games with star structure. Empirically, we show that although the initial belief heterogeneity vanishes in the limit, it plays a crucial role in equilibrium selection and helps select highly desirable equilibria.

## Acknowledgment

Shuyue Hu would like to thank Ms. Die Hu for her assistance in generating Figure 1 and the corresponding video presented in this paper. This research is supported in part by the National Research Foundation, Singapore and DSO National Laboratories under its AI Singapore Program (AISG Award No: AISG2-RP-2020-016), NRF2019-NRFANR095 ALIAS grant, NRF 2018 Fellowship NRF-NRFF2018-07, grant PIESGP-AI-2020-01, AME Programmatic Fund (Grant No.A20H6b0151) from A*STAR and Provost's Chair Professorship grant RGEPPV2101.

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

## A    Experiments on Network Competition

We have shown in Theorem 5 (of the main paper) that SFP converges to a unique QRE in any weighted zero-sum network population game even if there are multiple Nash equilibria underlying that game. In the following, we corroborate this by providing empirical evidence in agent-based simulations with different belief initialization.

**Game Description.**    Consider a five-population asymmetric matching pennies game [56], where the network structure is a line (depicted in Figure 1). Each agent has two actions $\{H, T\}$. Agents in populations 1 and 5 do not learn; they always play strategies $H$ and $T$, respectively. For agents in populations 2 to 4, they receive $+1$ if they match the strategy of the opponent in the next population, and receive $-1$ if they mismatch. On the contrary, they receive $+1$ if they mismatch the strategy of the opponent in the previous population, and receive $-1$ if they match. Hence, this game has infinitely many Nash equilibria of the form: agents in populations 2 and 4 play strategy $T$, whereas agents in population 3 are indifferent between strategies $H$ and $T$.

**Experimental Setups.**    In this game, agents in each population form two beliefs (one for the previous population and one for the next population). We are mainly interested in the strategies of population 3, as the Nash equilibria differ in the strategies in population 3. Thus, we let the initial beliefs about populations 1, 3 and 5 remain unchanged across different cases, and vary population 3's initial beliefs about populations 2 and 4. The initial beliefs about populations 1, 3 and 5, denoted by $\mu_{1H}$, $\mu_{3H}$ and $\mu_{5H}$, are distributed according to the distributions Beta$(20, 10)$, Beta$(6, 4)$, and Beta$(10, 5)$, respectively. The initial beliefs about populations are given in the legends of Figure 2. In all cases, the initial sum of weights $\lambda = 10$ and the temperature $\beta = 10$. Note that $\mu_{iT} = 1 - \mu_{iH}$ for all populations $i = 1, 2, 3, 4, 5$. We run 100 simulation runs for each initialization, and each simulation run consists of $1,000$ agents in each population.

**Results.**    As shown in Figure 4, given differential initialization of beliefs, agents in population 3 converge to the same equilibrium where they all take strategy $H$ with probability $0.5$. Therefore, even when the underlying zero-sum game has many Nash equilibria, SFP with different initial belief heterogeneity selects a unique equilibrium, addressing the problem of equilibrium selection.

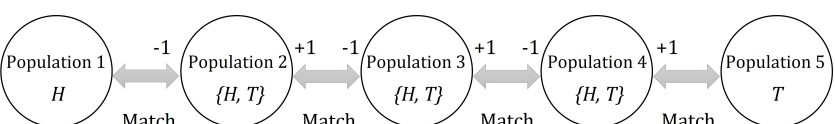

Figure 3: Asymmetric Matching Pennies.

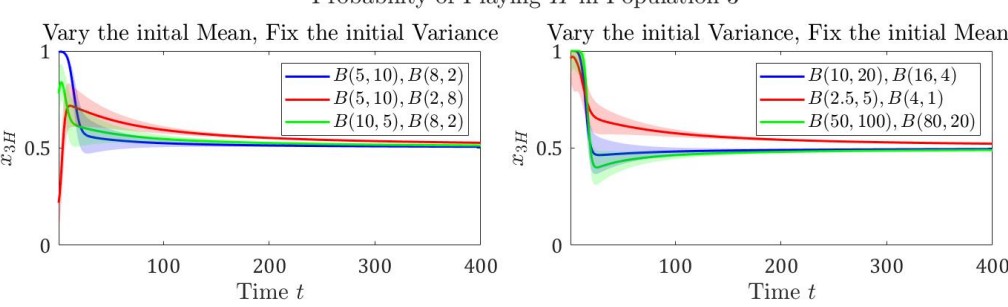

Figure 4: With different belief initialization, SFP selects a unique equilibrium where all agents in population 3 play strategy $H$ with probability $0.5$. The thin lines represent the mean mixed strategy (the choice probability of $H$) and the shaded areas represent the variance of the mixed strategies in the population. In the legends, $B$ denotes Beta distribution; the two Beta distributions correspond to the initial beliefs about the neighbor populations 2 and 4, respectively.

## B   Proof of Proposition 1

It follows from Equation 2 and Equation 3 of the main paper that the change in $\boldsymbol{\mu}_j^i(k,t)$ between two discrete time steps is

$$\boldsymbol{\mu}_j^i(k,t+1) = \boldsymbol{\mu}_j^i(k,t) + \frac{\bar{\mathbf{x}}_j(t) - \boldsymbol{\mu}_j^i(k,t)}{\lambda + t + 1}. \tag{14}$$

**Lemma 3.** *Under Assumption 1 of the main paper, for an arbitrary agent $k$ in population $i$, its belief $\boldsymbol{\mu}_j^i(k,t)$ about a neighbor population $j$ will never reach the extreme belief, i.e., the probability density for the boundary of the simplex $\Delta_i$ will remain zero.*

*Proof.* Assumption 1 ensures that $\bar{\mathbf{x}}_j(0)$ is in the interior of the simplex $\Delta_j$. Moreover, the logit choice function (Equation 5 in the main paper) also ensures that $\bar{\mathbf{x}}_j(t)$ stays in the interior of $\Delta_j$ afterwards for a finite temperature $\beta$. Hence, from Equation 14, one can see that $\boldsymbol{\mu}_j^i(k,t)$ for every time step $t$ will stay in the interior of $\Delta_j$. $\qquad\square$

In the following, for notation convenience, we sometimes drop the agent index $k$ and the time index $t$ depending on the context. Consider a population $i$. We rewrite the change in the beliefs about this population as follows.

$$\boldsymbol{\mu}_i(t+1) = \boldsymbol{\mu}_i(t) + \frac{\bar{\mathbf{x}}_i(t) - \boldsymbol{\mu}_i(t)}{\lambda + t + 1}. \tag{15}$$

Suppose that the amount of time that passes between two successive time steps is $\delta \in (0,1]$. We rewrite the above equation as

$$\boldsymbol{\mu}_i(t+\delta) = \boldsymbol{\mu}_i(t) + \delta \frac{\bar{\mathbf{x}}_i(t) - \boldsymbol{\mu}_i(t)}{\lambda + t + 1}. \tag{16}$$

Next, we consider a test function $\theta(\boldsymbol{\mu}_i)$. Define

$$Y = \frac{\mathbb{E}[\theta(\boldsymbol{\mu}_i(t+\delta))] - \mathbb{E}[\theta(\boldsymbol{\mu}_i(t))]}{\delta}. \tag{17}$$

Applying Taylor series for $\theta(\boldsymbol{\mu}_i(t+\delta))$ at $\boldsymbol{\mu}_i(t)$, we obtain

$$\theta(\boldsymbol{\mu}_i(t+\delta)) = \theta(\boldsymbol{\mu}_i(t)) + \frac{\delta}{\lambda + t + 1} \partial_{\boldsymbol{\mu}_i} \theta(\boldsymbol{\mu}_i) \left[\bar{\mathbf{x}}_i(t) - \boldsymbol{\mu}_i(t)\right]$$
$$+ \frac{\delta^2}{2(\lambda + t + 1)^2} \left[\bar{\mathbf{x}}_i(t) - \boldsymbol{\mu}_i(t)\right]^\top \mathbf{H}\theta(\boldsymbol{\mu}_i) \left[\bar{\mathbf{x}}_i(t) - \boldsymbol{\mu}_i(t)\right]$$
$$+ o\left(\left[\delta \frac{\bar{\mathbf{x}}_i(t) - \boldsymbol{\mu}_i(t)}{\lambda + t + 1}\right]^2\right) \tag{18}$$

where $\mathbf{H}$ denotes the Hessian matrix. Hence, the expectation $\mathbb{E}[\theta(\boldsymbol{\mu}_i(t+\delta))]$ is

$$\mathbb{E}[\theta(\boldsymbol{\mu}_i(t+\delta))] = \mathbb{E}[\theta(\boldsymbol{\mu}_i(t))] + \frac{\delta}{\lambda + t + 1} \mathbb{E}[\partial_{\boldsymbol{\mu}_i} \theta(\boldsymbol{\mu}_i(t))(\bar{\mathbf{x}}_i(t) - \boldsymbol{\mu}_i(t))]$$
$$+ \frac{\delta^2}{2(\lambda + t + 1)^2} \mathbb{E}\left[[\bar{\mathbf{x}}_i(t) - \boldsymbol{\mu}_i(t)]^\top \mathbf{H}\theta(\boldsymbol{\mu}_i) [\bar{\mathbf{x}}_i(t) - \boldsymbol{\mu}_i(t)]\right]$$
$$+ \frac{\delta^2}{2(\lambda + t + 1)^2} \mathbb{E}[o([\bar{\mathbf{x}}_i(t) - \boldsymbol{\mu}_i(t)]^2)] \tag{19}$$

Moving the term $\mathbb{E}[\theta(\boldsymbol{\mu}_i(t))]$ to the left hand side and dividing both sides by $\delta$, we recover the quantity $Y$, i.e.,

$$Y = \frac{1}{\lambda + t + 1} \mathbb{E}[\partial_{\boldsymbol{\mu}_i} \theta(\boldsymbol{\mu}_i(t))(\bar{\mathbf{x}}_i(t) - \boldsymbol{\mu}_i(t))]$$
$$+ \frac{\delta}{2(\lambda + t + 1)^2} \mathbb{E}[[\bar{\mathbf{x}}_i(t) - \boldsymbol{\mu}_i(t)]^\top \mathbf{H}\theta(\boldsymbol{\mu}_i(t))[\bar{\mathbf{x}}_i(t) - \boldsymbol{\mu}_i(t)] + o\left((\bar{\mathbf{x}}_i(t) - \boldsymbol{\mu}_i(t))^2\right)] \tag{20}$$

Taking the limit of $Y$ with $\delta \to 0$, the contribution of the second term on the right hand side vanishes, yielding

$$\lim_{\delta \to 0} Y = \frac{1}{\lambda + t + 1} \mathbb{E}[\partial_{\mu_i} \theta(\mu_i(t))(\bar{\mathbf{x}}_i(t) - \mu_i(t))] \tag{21}$$

$$= \frac{1}{\lambda + t + 1} \int p(\mu_i(t), t) \left[\partial_{\mu_i} \theta(\mu_i(t))(\bar{\mathbf{x}}_i(t) - \mu_i(t))\right] d\mu_i(t). \tag{22}$$

Apply integration by parts. We obtain

$$\lim_{\delta \to 0} Y = 0 - \frac{1}{\lambda + t + 1} \int \theta(\mu_i(t)) \nabla \cdot \left[p(\mu_i(t), t)(\bar{\mathbf{x}}_i(t) - \mu_i(t))\right] d\mu_i(t) \tag{23}$$

where we have leveraged that the probability mass $p(\mu_i, t)$ at the boundary $\partial \Delta_i$ remains zero as a result of Lemma 1. On the other hand, according to the definition of $Y$,

$$\lim_{\delta \to 0} Y = \lim_{\delta \to 0} \int \theta(\mu_i(t)) \frac{p(\mu_i, t + \delta) - p(\mu_i, t)}{\delta} d\mu_i = \int \theta(\mu_i(t)) \partial_t p(\mu_i, t) d\mu_i. \tag{24}$$

Therefore, we have the equality

$$\int \theta(\mu_i(t)) \partial_t p(\mu_i, t) d\mu_i = -\frac{1}{\lambda + t + 1} \int \theta(\mu_i(t)) \nabla \cdot \left[p(\mu_i(t), t)(\bar{\mathbf{x}}_i(t) - \mu_i(t))\right] d\mu_i(t). \tag{25}$$

As $\theta$ is a test function, this leads to

$$\partial_t p(\mu_i, t) = -\frac{1}{\lambda + t + 1} \nabla \cdot \left[p(\mu_i(t), t)(\bar{\mathbf{x}}_i(t) - \mu_i(t))\right]. \tag{26}$$

Rearranging the terms, we obtain Equation 7 of the main paper. By the definition of expectation given a probability distribution, it is straightforward to obtain Equation 8 of the main paper. Q.E.D.

## C  Proof of Theorem 1

Without loss of generality, we consider the variance of the belief $\mu_{is_i}$ about strategy $s_i$ of population $i$. Note that

$$\mathrm{Var}(\mu_{is_i}) = \mathbb{E}[(\mu_{is_i})^2] - (\bar{\mu}_{is_i})^2. \tag{27}$$

Hence, we have

$$\frac{d\mathrm{Var}(\mu_{is_i})}{dt} = \frac{d\mathbb{E}[(\mu_{is_i})^2]}{dt} - 2\bar{\mu}_{is_i} \frac{d\bar{\mu}_{is_i}}{dt}. \tag{28}$$

Consider the first term on the right hand side. We apply the Leibniz rule to interchange differentiation and integration, and then substitute $\frac{\partial p(\mu_i, t)}{\partial t}$ with Equation 8 in the main paper.

$$\frac{d\mathbb{E}[(\mu_{is_i})^2]}{dt}$$

$$= \int (\mu_{is_i})^2 \frac{\partial p(\mu_i, t)}{\partial t} d\mu_i \tag{29}$$

$$= -\int (\mu_{is_i})^2 \nabla \cdot \left(p(\mu_i, t) \frac{\bar{\mathbf{x}}_i - \mu_i}{\lambda + t + 1}\right) d\mu_i \tag{30}$$

$$= -\int (\mu_{is_i})^2 \sum_{s_i \in S_i} \partial_{\mu_{is_i}} \left(p(\mu_i, t) \frac{\bar{x}_{is_i} - \mu_{is_i}}{\lambda + t + 1}\right) d\mu_i \tag{31}$$

$$= \gamma \int (\mu_{is_i})^2 \sum_{s_i \in S_i} \partial_{\mu_{is_i}} p(\mu_i, t) \left(\bar{x}_{is_i} - \mu_{is_i}\right) d\mu_i + \gamma \int (\mu_{is_i})^2 p(\mu_i, t) \sum_{s_i \in S_i} \partial_{\mu_{is_i}} \left(\bar{x}_{is_i} - \mu_{is_i}\right) d\mu_i \tag{32}$$

where $\gamma := -\frac{1}{\lambda + t + 1}$. Applying integration by parts to the first term in Equation 32 yields

$$\int (\mu_{is_i})^2 \sum_{s_i \in S_i} \partial_{\mu_{is_i}} p(\mu_i, t) \left(\bar{x}_{is_i} - \mu_{is_i}\right) d\mu_i$$

$$= -\int (\mu_{is_i})^2 p(\mu_i, t) \left[\sum_{s_i' \in S_i} \partial_{\mu_{is_i'}} \left(\bar{x}_{is_i'} - \mu_{is_i'}\right)\right] + p(\mu_i, t) \partial_{\mu_{is_i}} \left[(\mu_{is_i})^2 \left(\bar{x}_{is_i} - \mu_{is_i}\right)\right] d\mu_i \tag{33}$$

where we have leveraged that the probability mass at the boundary remains zero (Lemma 1). Combining the above two equations, we obtain

$$\frac{d\mathbb{E}[(\mu_{is_i})^2]}{dt}$$

$$= -\gamma \int (\mu_{is_i})^2 p(\boldsymbol{\mu}_i, t) \left[ \sum_{s_i' \in S_i} \partial_{\mu_{is_i'}} (\bar{x}_{is_i'} - \mu_{is_i'}) \right] + p(\boldsymbol{\mu}_i, t) \partial_{\mu_{is_i}} \left[ (\mu_{is_i})^2 (\bar{x}_{is_i} - \mu_{is_i}) \right] d\boldsymbol{\mu}_i$$

$$+ \gamma \int (\mu_{is_i})^2 p(\boldsymbol{\mu}_i, t) \sum_{s_i \in S_i} \partial_{\mu_{is_i}} (\bar{x}_{is_i} - \mu_{is_i}) \, d\boldsymbol{\mu}_i \tag{34}$$

$$= \gamma \int \left[ -p(\boldsymbol{\mu}_i, t) \partial_{\mu_{is_i}} \left[ (\mu_{is_i})^2 (\bar{x}_{is_i} - \mu_{is_i}) \right] \right] + (\mu_{is_i})^2 p(\boldsymbol{\mu}_i, t) \partial_{\mu_{is_i}} (\bar{x}_{is_i} - \mu_{is_i}) \, d\boldsymbol{\mu}_i \tag{35}$$

$$= \gamma \int 2(\mu_{is_i})^2 p(\boldsymbol{\mu}_i, t) d\boldsymbol{\mu}_i - \gamma \int 2\bar{x}_{is_i} \mu_{is_i} p(\boldsymbol{\mu}_i, t) d\boldsymbol{\mu}_i \tag{36}$$

$$= -\frac{2\mathbb{E}[(\mu_{is_i})^2] - 2\bar{x}_{is_i} \bar{\mu}_{is_i}}{\lambda + t + 1}. \tag{37}$$

Next, we consider the second term in Equation 28. By Lemma 4, we have

$$2\bar{\mu}_{is_i} \frac{d\bar{\mu}_{is_i}}{dt} = \frac{2\bar{\mu}_{is_i}(\bar{x}_{is_i} - \bar{\mu}_{is_i})}{\lambda + t + 1}. \tag{38}$$

Combining Equations 37 and 38, the dynamics of the variance is

$$\frac{d\mathrm{Var}(\mu_{is_i})}{dt} = -\frac{2\mathbb{E}[(\mu_{is_i})^2] - 2\bar{x}_{is_i} \bar{\mu}_{is_i}}{\lambda + t + 1} - \frac{2\bar{\mu}_{is_i}(\bar{x}_{is_i} - \bar{\mu}_{is_i})}{\lambda + t + 1} \tag{39}$$

$$= \frac{2(\bar{\mu}_{is_i})^2 - 2\mathbb{E}[(\mu_{is_i})^2]}{\lambda + t + 1} \tag{40}$$

$$= -\frac{2\mathrm{Var}(\mu_{is_i})}{\lambda + t + 1}. \tag{41}$$

Q.E.D.

## D  Proof of Proposition 2

**Lemma 4.** *The dynamics of the mean belief* $\bar{\boldsymbol{\mu}}_i$ *about each population* $i \in V$ *is governed by a differential equation*

$$\frac{d\bar{\mu}_{is_i}}{dt} = \frac{\bar{x}_{is_i} - \bar{\mu}_{is_i}}{\lambda + t + 1}, \qquad \forall s_i \in S_i. \tag{42}$$

*Proof.* The time derivative of the mean belief about strategy $s_i$ is

$$\frac{d\bar{\mu}_{is_i}}{dt} = \frac{d}{dt} \int \mu_{is_i} p(\boldsymbol{\mu}_i, t) d\boldsymbol{\mu}_i. \tag{43}$$

We apply the Leibniz rule to interchange differentiation and integration, and then substitute $\frac{\partial p(\boldsymbol{\mu}_i, t)}{\partial t}$ with Equation 8 in the main paper.

$$\frac{d}{dt} \int \mu_{is_i} p(\boldsymbol{\mu}_i, t) d\boldsymbol{\mu}_i \tag{44}$$

$$= \int \mu_{is_i} \frac{\partial p(\boldsymbol{\mu}_i, t)}{\partial t} d\boldsymbol{\mu}_i \tag{45}$$

$$= -\int \mu_{is_i} \nabla \cdot \left( p(\boldsymbol{\mu}_i, t) \frac{\bar{\mathbf{x}}_i - \boldsymbol{\mu}_i}{\lambda + t + 1} \right) d\boldsymbol{\mu}_i \tag{46}$$

$$= -\int \mu_{is_i} \sum_{s_i \in S_i} \partial_{\mu_{is_i}} \left( p(\boldsymbol{\mu}_i, t) \frac{\bar{x}_{is_i} - \mu_{is_i}}{\lambda + t + 1} \right) d\boldsymbol{\mu}_i \tag{47}$$

$$= \gamma \left[ \int \mu_{is_i} \sum_{s_i \in S_i} \left( \partial_{\mu_{is_i}} p(\boldsymbol{\mu}_i, t) \right) (\bar{x}_{is_i} - \mu_{is_i}) d\boldsymbol{\mu}_i + \int \mu_{is_i} p(\boldsymbol{\mu}_i, t) \sum_{s_i \in S_i} \partial_{\mu_{is_i}} (\bar{x}_{is_i} - \mu_{is_i}) d\boldsymbol{\mu}_i \right] \tag{48}$$

where $\gamma := -\frac{1}{\lambda + t + 1}$. Apply integration by parts to the first term in Equation 48.

$$\int \mu_{is_i} \sum_{s_i \in S_i} \left( \partial_{\mu_{is_i}} p(\boldsymbol{\mu}_i, t) \right) (\bar{x}_{is_i} - \mu_{is_i}) d\boldsymbol{\mu}_i$$

$$= -\int \mu_{is_i} p(\boldsymbol{\mu}_i, t) \left[ \sum_{s_i' \in S_i} \partial_{\mu_{is_i'}} (\bar{x}_{is_i'} - \mu_{is_i'}) \right] + p(\boldsymbol{\mu}_i, t) \partial_{\mu_{is_i}} \left[ \mu_{is_i} (\bar{x}_{is_i} - \mu_{is_i}) \right] d\boldsymbol{\mu}_i \tag{49}$$

where we have leveraged that the probability mass at the boundary remains zero. Hence, it follows from Equation 48 that

$$\frac{d}{dt} \int \mu_{is_i} p(\boldsymbol{\mu}_i, t) d\boldsymbol{\mu}_i \tag{50}$$

$$= -\gamma \int \mu_{is_i} p(\boldsymbol{\mu}_i, t) \sum_{s_i' \in S_i} \partial_{\mu_{is_i'}} (\bar{x}_{is_i'} - \mu_{is_i'}) d\boldsymbol{\mu}_i - \gamma \int p(\boldsymbol{\mu}_i, t) \partial_{\mu_{is_i}} \left[ \mu_{is_i} (\bar{x}_{is_i} - \mu_{is_i}) \right] d\boldsymbol{\mu}_i$$

$$+ \gamma \int \mu_{is_i} p(\boldsymbol{\mu}_i, t) \sum_{s_i \in S_i} \partial_{\mu_{is_i}} (\bar{x}_{is_i} - \mu_{is_i}) d\boldsymbol{\mu}_i \tag{51}$$

$$= \gamma \int p(\boldsymbol{\mu}_i, t) \left[ \mu_{is_i} \partial_{\mu_{is_i}} (\bar{x}_{is_i} - \mu_{is_i}) - \partial_{\mu_{is_i}} \left[ \mu_{is_i} (\bar{x}_{is_i} - \mu_{is_i}) \right] \right] d\boldsymbol{\mu}_i \tag{52}$$

$$= \gamma \int p(\boldsymbol{\mu}_i, t) \mu_{is_i} d\boldsymbol{\mu}_i - \int p(\boldsymbol{\mu}_i, t) \bar{x}_{is_i} d\boldsymbol{\mu}_i \tag{53}$$

$$= \frac{\bar{x}_{is_i} - \bar{\mu}_{is_i}}{\lambda + t + 1} \tag{54}$$

$\square$

We repeat the mean probability $\bar{x}_{is_i}$, which has been given in Equation 8 in the main paper, as follows:

$$\bar{x}_{is_i} = \int \frac{\exp(\beta u_{is_i})}{\sum_{s_i' \in S_i} \exp(\beta u_{is_i'})} \prod_{j \in V_i} p(\boldsymbol{\mu}_j, t) \left( \prod_{j \in V_i} d\boldsymbol{\mu}_j \right) \tag{55}$$

where $u_{is_i} = \sum_{j \in V_i} \mathbf{e}_{s_i}^\top \mathbf{A}_{ij} \boldsymbol{\mu}_j$. Define $\bar{\boldsymbol{\mu}} := \{\bar{\boldsymbol{\mu}}_j\}_{j \in V_i}$ and

$$f_{s_i}(\{\boldsymbol{\mu}_j\}_{j \in V_i}) := \frac{\exp\left(\beta \sum_{j \in V_i} \mathbf{e}_{s_i}^\top \mathbf{A}_{ij} \boldsymbol{\mu}_j\right)}{\sum_{s_i' \in S_i} \exp\left(\beta \sum_{j \in V_i} \mathbf{e}_{s_i'}^\top \mathbf{A}_{ij} \boldsymbol{\mu}_j\right)}. \tag{56}$$

Applying the Taylor expansion to approximate this function at the mean belief $\bar{\boldsymbol{\mu}}$, we have

$$f_{s_i}(\{\boldsymbol{\mu}_j\}_{j \in V_i}) = f_{s_i}(\bar{\boldsymbol{\mu}}) + \nabla f_{s_i}(\bar{\boldsymbol{\mu}}) \cdot (\boldsymbol{\mu} - \bar{\boldsymbol{\mu}}) + \frac{1}{2!}(\boldsymbol{\mu} - \bar{\boldsymbol{\mu}})^\top \mathbf{H} f_{s_i}(\bar{\boldsymbol{\mu}})(\boldsymbol{\mu} - \bar{\boldsymbol{\mu}}) + \frac{1}{3!}O(||\boldsymbol{\mu} - \bar{\boldsymbol{\mu}}||^3) \tag{57}$$

where $\mathbf{H}$ denotes the Hessian matrix. Hence, we can rewrite Equation 55 as

$$\bar{x}_{is_i} = \int f_{s_i}(\{\boldsymbol{\mu}_j\}_{j \in V_i}) \prod_{j \in V_i} p(\boldsymbol{\mu}_j, t) \left( \prod_{j \in V_i} d\boldsymbol{\mu}_j \right) \tag{58}$$

$$\approx f_{s_i}(\bar{\boldsymbol{\mu}}) + \int \nabla f_{s_i}(\bar{\boldsymbol{\mu}}) \cdot \boldsymbol{\mu} \prod_{j \in V_i} p(\boldsymbol{\mu}_j, t) \left( \prod_{j \in V_i} d\boldsymbol{\mu}_j \right) - \nabla f_{s_i}(\bar{\boldsymbol{\mu}}) \cdot \bar{\boldsymbol{\mu}}$$

$$+ \int \frac{1}{2}(\boldsymbol{\mu} - \bar{\boldsymbol{\mu}})^\top \mathbf{H} f_{s_i}(\bar{\boldsymbol{\mu}})(\boldsymbol{\mu} - \bar{\boldsymbol{\mu}}) \prod_{j \in V_i} p(\boldsymbol{\mu}_j, t) \left( \prod_{j \in V_i} d\boldsymbol{\mu}_j \right)$$

$$+ \int \frac{1}{3!}O(||\boldsymbol{\mu} - \bar{\boldsymbol{\mu}}||)^3 \prod_{j \in V_i} p(\boldsymbol{\mu}_j, t) \left( \prod_{j \in V_i} d\boldsymbol{\mu}_j \right) \tag{59}$$

Observe that in Equation 59, the second and the third term can be canceled out. Moreover, for any two neighbor populations $j, k \in V_i$, the beliefs $\boldsymbol{\mu}_j, \boldsymbol{\mu}_k$ about these two populations are updated separately and independently. Hence, the covariance of these beliefs are zero. We apply the moment closure approximation [64, 39] with the second order and obtain

$$\bar{x}_{is_i} \approx f_{s_i}(\bar{\boldsymbol{\mu}}) + \frac{1}{2} \sum_{j \in V_i} \sum_{s_j \in S_j} \frac{\partial^2 f_{s_i}(\bar{\boldsymbol{\mu}})}{(\partial \mu_{js_j})^2} \mathrm{Var}(\mu_{js_j}). \tag{60}$$

Hence, substituting $\bar{x}_{is_i}$ in Lemma 4 with the above approximation, we have the mean belief dynamics

$$\frac{d\bar{\mu}_{is_i}}{dt} \approx \frac{f_{s_i}(\bar{\boldsymbol{\mu}}) - \bar{\mu}_{is_i}}{\lambda + t + 1} + \frac{\sum_{j \in V_i} \sum_{s_j \in S_j} \frac{\partial^2 f_{s_i}(\bar{\boldsymbol{\mu}})}{(\partial \mu_{js_j})^2} \mathrm{Var}(\mu_{js_j})}{2(\lambda + t + 1)}. \tag{61}$$

Q.E.D.

# E   Proof of Proposition 3

It follows from Equation 2 and Equation 3 of the main paper that the change in beliefs between two successive time steps is as follows.

$$\boldsymbol{\mu}_i(t+1) = \boldsymbol{\mu}_i(t) + \frac{\mathbf{x}_i(t) - \boldsymbol{\mu}_i(t)}{\lambda + t + 1}. \tag{62}$$

Suppose that the amount of time that passes between two successive time steps is $\delta \in (0, 1]$. We rewrite the above equation as

$$\boldsymbol{\mu}_i(t+\delta) = \boldsymbol{\mu}_i(t) + \delta \frac{\mathbf{x}_i(t) - \boldsymbol{\mu}_i(t)}{\lambda + t + 1}. \tag{63}$$

Move the term $\boldsymbol{\mu}_i(t)$ to the right hand side and divide both sides by $\delta$,

$$\frac{\boldsymbol{\mu}_i(t+\delta) - \boldsymbol{\mu}_i(t)}{\delta} = \frac{\mathbf{x}_i(t) - \boldsymbol{\mu}_i(t)}{\lambda + t + 1}. \tag{64}$$

Assume that the amount of time $\delta$ between two successive time steps goes to zero. we have

$$\frac{d\boldsymbol{\mu}_i}{dt} = \lim_{\delta \to 0} \frac{\boldsymbol{\mu}_i(t+\delta) - \boldsymbol{\mu}_i(t)}{\delta} = \frac{\mathbf{x}_i(t) - \boldsymbol{\mu}_i(t)}{\lambda + t + 1}. \tag{65}$$

Q.E.D.

# F  Proof of Proposition 4

It is straightforward to see that

$$\frac{d\boldsymbol{\mu}_i}{dt} = \frac{\mathbf{x}_i - \boldsymbol{\mu}_i}{\lambda + t + 1} = 0 \implies \mathbf{x}_i = \boldsymbol{\mu}_i. \tag{66}$$

Denote the equilibrium points of the system dynamics, which satisfies the above equation, by $(\mathbf{x}_i^*, \boldsymbol{\mu}_i^*)$ for each population $i$. By the logit choice rule, we have

$$x_{is_i}^* = \frac{\exp\left(\beta u_{is_i}\right)}{\sum_{s_i' \in S_i} \exp\left(\beta u_{is_i'}\right)} = \frac{\exp\left(\beta \sum_{j \in V_i} \mathbf{e}_{s_i}^\top \mathbf{A}_{ij} \boldsymbol{\mu}_j^*\right)}{\sum_{s_i' \in S_i} \exp\left(\beta \sum_{j \in V_i} \mathbf{e}_{s_i'}^\top \mathbf{A}_{ij} \boldsymbol{\mu}_j^*\right)}. \tag{67}$$

Leveraging that $\mathbf{x}_i^* = \boldsymbol{\mu}_i^*, \forall i \in V$ at equilibrium, we can replace $\boldsymbol{\mu}_j^*$ with $\mathbf{x}_j^*$. Q.E.D.

# G  Proof of Theorem 2

Consider an agent $i$ in a classic network game. The set of neighbors is $V_i$, the set of beliefs about the neighbors is $\{\boldsymbol{\mu}_j\}_{j \in V_i}$, and the choice distribution is $\mathbf{x}_i$. Given a classic network game, the expected payoff is given by $\mathbf{x}_i^\top \sum_{(i,j) \in E} A_{ij} \boldsymbol{\mu}_j$. Define a perturbed payoff function

$$\pi_i\left(\mathbf{x}_i, \{\boldsymbol{\mu}_j\}_{j \in V_i}\right) := \mathbf{x}_i^\top \sum_{j \in V_i} A_{ij} \boldsymbol{\mu}_j + v(\mathbf{x}_i) \tag{68}$$

where $v(\mathbf{x}_i) = -\frac{1}{\beta} \sum_{s_i \in S_i} x_{is_i} \ln(x_{is_i})$. Under this form of $v(\mathbf{x}_i)$, the maximization of $\pi_i$ yields the choice distribution $\mathbf{x}_i$ from the logit choice function [33]. Based on this, we establish the following lemma.

**Lemma 5.** *For a choice distribution $\mathbf{x}_i$ of SFP in a network game,*

$$\partial_{\mathbf{x}_i} \pi_i\left(\mathbf{x}_i, \{\boldsymbol{\mu}_j\}_{j \in V_i}\right) = \mathbf{0} \quad \text{and} \quad \sum_{j \in V_i} (A_{ij} \boldsymbol{\mu}_j)^\top = -\partial_{\mathbf{x}_i} v(\mathbf{x}_i). \tag{69}$$

*Proof.* This lemma immediately follows from the fact that the maximization of $\pi_i$ will yield the choice distribution $\mathbf{x}_i$ from the logit choice function [33]. $\square$

The belief dynamics of an agent can be simplified after time-reparameterization.

**Lemma 6.** *Given $\tau = \ln \frac{\lambda+t+1}{\lambda+1}$, the belief dynamics of homogeneous systems (given in Equation 11 in the main paper) is equivalent to*

$$\frac{d\boldsymbol{\mu}_i}{d\tau} = \mathbf{x}_i - \boldsymbol{\mu}_i. \tag{70}$$

*Proof.* From $\tau = \ln \frac{\lambda+t+1}{\lambda+1}$, we have

$$t = (\lambda + 1)(\exp(\tau) - 1). \tag{71}$$

By the chain rule, for each dimension $s_i$,

$$\frac{d\mu_{is_i}}{d\tau} = \frac{d\mu_{is_i}}{dt} \frac{dt}{d\tau} \tag{72}$$

$$= \frac{x_{is_i} - \mu_{is_i}}{\lambda + t + 1} \frac{d\left((\lambda+1)(\exp(\tau) - 1)\right)}{d\tau} \tag{73}$$

$$= \frac{x_{is_i} - \mu_{is_i}}{\lambda + (\lambda+1)(\exp(\tau) - 1) + 1} (\lambda + 1) \exp(\tau) \tag{74}$$

$$= x_{is_i} - \mu_{is_i}. \tag{75}$$

$\square$

Next, we define the Lyapunov function $L$ as

$$L := \sum_{i \in V} \omega_i L_i \quad \text{s.t.} \quad L_i := \pi_i \left( \mathbf{x}_i, \{\boldsymbol{\mu}_j\}_{j \in V_i} \right) - \pi_i \left( \boldsymbol{\mu}_i, \{\boldsymbol{\mu}_j\}_{j \in V_i} \right). \tag{76}$$

where $\{\omega_i\}_{i \in V}$ is the set of positive weights defined in the weighted zero-sum $\Gamma$. The function $L$ is non-negative because for every $i \in V$, $\mathbf{x}_i$ maximizes the function $\pi_i$. When for every $i \in V$, $\mathbf{x}_i = \boldsymbol{\mu}_i$, the function $L$ reaches the minimum value 0.

Rewrite $L$ as

$$L = \sum_{i \in V} \left[ \omega_i \pi_i \left( \mathbf{x}_i, \{\boldsymbol{\mu}_j\}_{j \in V_i} \right) - \omega_i \boldsymbol{\mu}_i^\top \sum_{j \in V_i} A_{ij} \boldsymbol{\mu}_j - \omega_i v(\boldsymbol{\mu}_i) \right]. \tag{77}$$

We observe that $\pi_i \left( \mathbf{x}_i, \{\boldsymbol{\mu}_j\}_{j \in V_i} \right)$ is convex in $\boldsymbol{\mu}_j, j \in V_i$ by Danskin's theorem, and $-v(\boldsymbol{\mu}_i)$ is strictly convex in $\boldsymbol{\mu}_i$. Moreover, by the weighted zero-sum property given in Equation 2 in the main paper, we have

$$\sum_{i \in V} \left( \omega_i \boldsymbol{\mu}_i^\top \sum_{j \in V_i} A_{ij} \boldsymbol{\mu}_j \right) = 0 \tag{78}$$

since $\mu_i \in \Delta_i, \mu_j \in \Delta_j$ for every $i, j \in V$. Therefore, the function $L$ is a strictly convex function and attains its minimum value 0 at a unique point $\mathbf{x}_i = \boldsymbol{\mu}_i, \forall i \in V$.

Consider the function $L_i$. Its time derivative is

$$\dot{L}_i = \partial_{\mathbf{x}_i} \pi_i \left( \mathbf{x}_i, \{\boldsymbol{\mu}_j\}_{j \in V_i} \right) \dot{\mathbf{x}}_i + \sum_{j \in V_i} \left[ \partial_{\boldsymbol{\mu}_j} \pi_i \left( \mathbf{x}_i, \{\boldsymbol{\mu}_j\}_{j \in V_i} \right) \dot{\boldsymbol{\mu}}_j \right]$$
$$- \partial_{\boldsymbol{\mu}_i} \pi_i \left( \boldsymbol{\mu}_i, \{\boldsymbol{\mu}_j\}_{j \in V_i} \right) \dot{\boldsymbol{\mu}}_i - \sum_{j \in V_i} \left[ \partial_{\boldsymbol{\mu}_j} \pi_i \left( \boldsymbol{\mu}_i, \{\boldsymbol{\mu}_j\}_{j \in V_i} \right) \dot{\boldsymbol{\mu}}_j \right]. \tag{79}$$

Note that the partial derivative $\partial_{\mathbf{x}_i} \pi_i$ equals $\mathbf{0}$ by Lemma 5. Thus, we can rewrite this as

$$\dot{L}_i = \partial_{\boldsymbol{\mu}_i} \pi_i \left( \boldsymbol{\mu}_i, \{\boldsymbol{\mu}_j\}_{j \in V_i} \right) \dot{\boldsymbol{\mu}}_i + \sum_{j \in V_i} \left[ \partial_{\boldsymbol{\mu}_j} \pi_i \left( \mathbf{x}_i, \{\boldsymbol{\mu}_j\}_{j \in V_i} \right) - \partial_{\boldsymbol{\mu}_j} \pi_i \left( \boldsymbol{\mu}_i, \{\boldsymbol{\mu}_j\}_{j \in V_i} \right) \right] \dot{\boldsymbol{\mu}}_j \tag{80}$$

$$= - \left[ \sum_{j \in V_i} \left( A_{ij} \boldsymbol{\mu}_j \right)^\top + \partial_{\boldsymbol{\mu}_i} v(\boldsymbol{\mu}_i) \right] (\mathbf{x}_i - \boldsymbol{\mu}_i) + \sum_{j \in V_i} \left( \mathbf{x}_i^\top A_{ij} - \boldsymbol{\mu}_i^\top A_{ij} \right) (\mathbf{x}_j - \boldsymbol{\mu}_j) \tag{81}$$

$$= \left[ \partial_{\mathbf{x}_i} v(\mathbf{x}_i) - \partial_{\boldsymbol{\mu}_i} v(\boldsymbol{\mu}_i) \right] (\mathbf{x}_i - \boldsymbol{\mu}_i) + \sum_{j \in V_i} \left( \mathbf{x}_i^\top A_{ij} \mathbf{x}_j - \boldsymbol{\mu}_i^\top A_{ij} \mathbf{x}_j - \mathbf{x}_i^\top A_{ij} \boldsymbol{\mu}_j + \boldsymbol{\mu}_i^\top A_{ij} \boldsymbol{\mu}_j \right). \tag{82}$$

where from Equation 81 to 82, we apply Lemma 5 to substitute $\sum_{j \in V_i} \left( A_{ij} \boldsymbol{\mu}_j \right)^\top$ with $-\partial_{\mathbf{x}_i} v(\mathbf{x}_i)$. Hence, summing over all the populations, the time derivative of $L$ is

$$\dot{L} = \sum_{i \in V} \omega_i \left[ \partial_{\mathbf{x}_i} v(\mathbf{x}_i) - \partial_{\boldsymbol{\mu}_i} v(\boldsymbol{\mu}_i) \right] (\mathbf{x}_i - \boldsymbol{\mu}_i)$$
$$+ \sum_{i \in V} \sum_{j \in V_i} \omega_i \left( \mathbf{x}_i^\top A_{ij} \mathbf{x}_j - \boldsymbol{\mu}_i^\top A_{ij} \mathbf{x}_j - \mathbf{x}_i^\top A_{ij} \boldsymbol{\mu}_j + \boldsymbol{\mu}_i^\top A_{ij} \boldsymbol{\mu}_j \right). \tag{83}$$

The summation in the second line is equivalent to

$$\sum_{(i,j) \in E} \left( \omega_i \mathbf{x}_i^\top A_{ij} \mathbf{x}_j + \omega_j \mathbf{x}_j^\top A_{ji} \mathbf{x}_i \right) - \left( \omega_i \boldsymbol{\mu}_i^\top A_{ij} \mathbf{x}_j + \omega_j \mathbf{x}_j^\top A_{ji} \boldsymbol{\mu}_i \right) \tag{84}$$

$$- \left( \omega_i \mathbf{x}_i^\top A_{ij} \boldsymbol{\mu}_j + \omega_j \boldsymbol{\mu}_j^\top A_{ji} \mathbf{x}_i \right) + \left( \omega_i \boldsymbol{\mu}_i^\top A_{ij} \boldsymbol{\mu}_j + \omega_j \boldsymbol{\mu}_j^\top A_{ji} \boldsymbol{\mu}_i \right). \tag{85}$$

By the weighted zero-sum property given in Equation 2 in the main paper, this summation equals 0, yielding

$$\dot{L} = \sum_{i \in V} \omega_i \left[ \partial_{\mathbf{x}_i} v(\mathbf{x}_i) - \partial_{\boldsymbol{\mu}_i} v(\boldsymbol{\mu}_i) \right] (\mathbf{x}_i - \boldsymbol{\mu}_i). \tag{86}$$

Note that the function $v$ is strictly concave such that its second derivative is negative definite. By this property, $\dot{L} \leq 0$ with equality only if $\mathbf{x}_i = \boldsymbol{\mu}_i, \forall i \in V$, which corresponds to the QRE. Therefore, $L$ is a strict Lyapunov function, and the global asymptotic stability of the QRE follows. Q.E.D.

# H  Proof of Theorem 3

Consider a root agent $j$ of a star structure. Its set of leaf (neighbor) agents is $V_j$, the set of beliefs about the leaf agents is $\{\mu_i\}_{i \in V_j}$, and the choice distribution is $\mathbf{x}_j$. Given the game $\Gamma$, the expected payoff is $\mathbf{x}_j^\top \sum_{i \in V_j} A_{ji}\mu_i$. Define a perturbed payoff function

$$\pi_j\left(\mathbf{x}_j, \{\mu_i\}_{i \in V_j}\right) := \mathbf{x}_j^\top \sum_{i \in V_j} A_{ji}\mu_i + v(\mathbf{x}_j) \tag{87}$$

where $v(\mathbf{x}_j) = -\frac{1}{\beta} \sum_{s_j \in S_j} x_{js_j} \ln(x_{js_j})$. Under this form of $v(\mathbf{x}_j)$, the maximization of $\pi_j$ yields the choice distribution $\mathbf{x}_j$ from the logit choice function [33].

Consider a leaf agent $i$ of the root agent $j$. It has only one neighbor, which is population $j$. Thus, given the game $\Gamma$, the expected payoff is $\mathbf{x}_i^\top A_{ij}\mu_j$. Define a perturbed payoff function

$$\pi_i\left(\mathbf{x}_i, \mu_j\right) := \mathbf{x}_i^\top A_{ij}\mu_j + v(\mathbf{x}_i) \tag{88}$$

where $v(\mathbf{x}_i) = -\frac{1}{\beta} \sum_{s_i \in S_i} x_{is_i} \ln(x_{is_i})$. Similarly, the maximization of $\pi_i$ yields the choice distribution $\mathbf{x}_i$ from the logit choice function [33]. Based on this, we establish the following lemma.

**Lemma 7.** *For choice distributions of SFP in a network game with a star structure,*

$$\partial_{\mathbf{x}_j}\pi_j\left(\mathbf{x}_j, \{\mu_i\}_{i \in V_j}\right) = \mathbf{0} \quad and \quad \sum_{i \in V_j}(A_{ji}\mu_i)^\top = -\partial_{\mathbf{x}_j}v(\mathbf{x}_j) \qquad \textit{if } j \textit{ is a root agent,} \tag{89}$$

$$\partial_{\mathbf{x}_i}\pi_i\left(\mathbf{x}_i, \mu_j\right) = \mathbf{0} \quad and \quad (A_{ij}\mu_j)^\top = -\partial_{\mathbf{x}_i}v(\mathbf{x}_i) \qquad \textit{if } i \textit{ is a leaf agent.} \tag{90}$$

*Proof.* This lemma immediately follows from the fact that the maximization of $\pi_j$ and $\pi_i$, respectively, yield the choice distributions $\mathbf{x}_j$ and $\mathbf{x}_i$ from the logit choice function [33]. $\square$

Let $\mathcal{R} \subset V$ be the set of all root agents. We define

$$L := \sum_{j \in \mathcal{R}} L_j \quad \text{s.t.} \quad L_j := \mu_j^\top \sum_{i \in V_j} A_{ji}\mu_i + v(\mu_j) + \sum_{i \in V_j} v(\mu_i). \tag{91}$$

Consider the function $L_j$. Its time derivative $\dot{L}_j$ is

$$\dot{L}_j = \left[\partial_{\mu_j}(\mu_j^\top \sum_{i \in V_j} A_{ji}\mu_i)\dot{\mu}_j + \sum_{i \in V_j} \partial_{\mu_i}(\mu_j^\top \sum_{i \in V_j} A_{ji}\mu_i)\dot{\mu}_i\right] + \partial_{\mu_j}v(\mu_j)\dot{\mu}_j + \sum_{i \in V_j} \partial_{\mu_i}v(\mu_i)\dot{\mu}_i \tag{92}$$

$$= \sum_{i \in V_j}(A_{ji}\mu_i)^\top(\mathbf{x}_j - \mu_j) + \left[\sum_{i \in V_j} \mu_j^\top A_{ji}(\mathbf{x}_i - \mu_i)\right] + \partial_{\mu_j}v(\mu_j)(\mathbf{x}_j - \mu_j) + \sum_{i \in V_j} \partial_{\mu_i}v(\mu_i)(\mathbf{x}_i - \mu_i). \tag{93}$$

Since we have $(A_{ij}\mu_j)^\top = \mu_j^\top A_{ij}^\top = \mu_j^\top A_{ji}$, applying Lemma 7, we can substitute $\sum_{i \in V_j}(A_{ji}\mu_i)^\top$ with $-\partial_{\mathbf{x}_j}v(\mathbf{x}_j)$, and $\mu_j^\top A_{ji}$ with $-\partial_{\mathbf{x}_i}v(\mathbf{x}_i)$, yielding

$$\dot{L}_j = -\partial_{\mathbf{x}_j}v(\mathbf{x}_j)(\mathbf{x}_j - \mu_j) + \left[\sum_{i \in V_j}(-\partial_{\mathbf{x}_i}v(\mathbf{x}_i))(\mathbf{x}_i - \mu_i)\right] + \partial_{\mu_j}v(\mu_j)(\mathbf{x}_j - \mu_j)$$

$$+ \sum_{i \in V_j} \partial_{\mu_i}v(\mu_i)(\mathbf{x}_i - \mu_i) \tag{94}$$

$$= (\partial_{\mu_j}v(\mu_j) - \partial_{\mathbf{x}_j}v(\mathbf{x}_j))(\mathbf{x}_j - \mu_j) + \sum_{i \in V_j}(\partial_{\mu_i}v(\mu_i) - \partial_{\mathbf{x}_i}v(\mathbf{x}_i))(\mathbf{x}_i - \mu_i) \tag{95}$$

Note that the function $v$ is strictly concave such that its second derivative is negative definite. By this property, $\dot{L}_j \geq 0$ with equality only if $\mathbf{x}_i = \mu_i, \forall i \in V_j$ and $\mathbf{x}_j = \mu_j$. Thus, the time derivative of the function $L$, i.e., $\dot{L} = \sum_{j \in \mathcal{R}} \dot{L}_j \geq 0$ with equality only if $\mathbf{x}_i = \mu_i, \forall i \in V_j, \mathbf{x}_j = \mu_j, \forall j \in \mathcal{R}$. Q.E.D.

# I   Proof of Lemma 1

**Definition 1.** *A nonautonomous system of differential equations in $R^n$*

$$x' = f(t, x) \tag{96}$$

*is said to be asymptotically autonomous with limit equation*

$$y' = g(y), \tag{97}$$

*if $f(t,x) \to g(x), t \to \infty$, where the convergence is uniform on each compact subset of $R^n$. Conventionally, the solution flow of Eq. 96 is called the asymptotically autonomous semiflow (denoted by $\phi$) and the solution flow of Eq. 97 is called the limit semiflow (denoted by $\Theta$).*

We first time-reparameterize the mean belief dynamics of heterogeneous systems. Assume $\tau = \ln \frac{\lambda+t+1}{\lambda+1}$. By the chain rule and Equation 61, for each dimension $s_i$,

$$\frac{d\bar{\mu}_{is_i}}{d\tau} = \frac{d\bar{\mu}_{is_i}}{dt} \frac{dt}{d\tau} \tag{98}$$

$$= \left[ \frac{f_{s_i}(\bar{\boldsymbol{\mu}}) - \bar{\mu}_{is_i}}{\lambda + t + 1} + \frac{\sum_{j \in V_i} \sum_{s_j \in S_j} \frac{\partial^2 f_{s_i}(\bar{\boldsymbol{\mu}})}{(\partial \mu_{js_j})^2} \mathrm{Var}(\mu_{js_j})}{2(\lambda + t + 1)} \right] \frac{d\left((\lambda+1)(\exp(\tau) - 1)\right)}{d\tau} \tag{99}$$

$$= \frac{f_{s_i}(\bar{\boldsymbol{\mu}}) - \bar{\mu}_{is_i} + \frac{1}{2} \sum_{j \in V_i} \sum_{s_j \in S_j} \frac{\partial^2 f_{s_i}(\bar{\boldsymbol{\mu}})}{(\partial \mu_{js_j})^2} \left(\frac{\lambda+1}{\lambda+t+1}\right)^2 \sigma^2(\mu_{js_j})}{\lambda + (\lambda+1)(\exp(\tau) - 1) + 1} (\lambda + 1) \exp(\tau) \tag{100}$$

$$= f_{s_i}(\bar{\boldsymbol{\mu}}) - \bar{\mu}_{is_i} + \frac{1}{2} \sum_{j \in V_i} \sum_{s_j \in S_j} \frac{\partial^2 f_{s_i}(\bar{\boldsymbol{\mu}})}{(\partial \mu_{js_j})^2} \sigma^2(\mu_{js_j}) \exp(-2\tau). \tag{101}$$

Observe that $\exp(-2\tau)$ decays to zero exponentially fast and that both $\sigma^2(\mu_{js_j})$ and $\frac{\partial^2 f_{s_i}(\bar{\boldsymbol{\mu}})}{(\partial \mu_{js_j})^2}$ are bounded for every $\boldsymbol{\mu}$ in the simplex $\prod_{j \in V_i} \Delta_j$. Hence, Equation 101 converges locally and uniformly to the following equation:

$$\frac{d\bar{\mu}_{is_i}}{d\tau} = f_{s_i}(\bar{\boldsymbol{\mu}}) - \bar{\mu}_{is_i}. \tag{102}$$

Note that $x_{is_i} = f_{s_i}(\bar{\boldsymbol{\mu}})$ for a single representative agent, and thus the above equation is algebraically equivalent to the limit equation in Lemma 1 of the main paper. Q.E.D.

## J   Numerical Methods, Source Code, and Computing Resource

**Numerical Method for the PDE model.**   Only limited types of PDEs allow analytic solutions. Hence, we numerically solve the PDE using the finite difference method [80]. The theoretical predictions in Figure 1 of the main paper are generated using the finite difference method given a specific initial setting (the initial sum of weights is $\lambda = 10$, the temperature is $\beta = 10$, the initial belief distribution is specified in the caption of the figures).

**Source Code and Computing Resource.**   The source code for reproducing our main experiments is included in the supplementary material (available at *https://sites.google.com/view/shuyue-hu*). The Matlab script *finitedifference.m* numerically solves our continuity equation model. Figure 1 of the main paper is generated using the Origins Lab based on the numerical solution of the continuity equations. The Matlab script *regionofattraction.m* visualizes the region of attraction of different equilibria in stag hunt games, which are depicted in Figure 2. The Python scripts *simulation(staghunt).py* and *simulation(matchingpennies).py* correspond to the agent-based simulations in two-population stag hunt games and five-population asymmetric matching pennies games, respectively. We use a laptop (CPU: AMD Ryzen 7 5800H) to run all the experiments.

