# OpenReview forum: "The Best of Both Worlds in Network Population Games: Reaching Consensus and Convergence to Equilibrium"
_NeurIPS.cc/2023/Conference — NeurIPS 2023 poster_

### Official Review · Reviewer_YUjL · 2023-07-03

**Soundness:** 4 excellent
**Presentation:** 3 good
**Contribution:** 4 excellent
**Rating:** 8
**Confidence:** 4

**Summary:**

This paper examines the connection between the notions of consensus and equilibrium in a multi-agent system where multiple interacting sub-populations coexist. They argue that consensus can be seen as an intricate component of intra-population stability, whereas equilibrium can be seen as encoding inter-population stability. They show that smooth fictitious play can achieve both consensus and convergence to equilibrium in diverse multi-agent settings.

**Strengths:**

Excellent paper that brings together the concepts of consensus and equilibrium. Strong theory, interesting experiments.

**Weaknesses:**

1) don't forget the conclusion in the final version of the paper.
2) It is clear that in the long run, due to the strong law of large numbers, FP is such that agents within a population will form the same beliefs. Therefore, the point of view taken by the authors is to study a representative agent. I say: what could be interesting is the transient regime where agents beliefs have not yet converged. There, one could use the central limit theorem, and look at the interplay between, on the one side, consensus that has not been reached yet, and the convergence of populations to equilibria. You could also look at it with 2 different learning rates (one for intra, one for inter populations, I suggest you explore connections with this https://arxiv.org/pdf/2205.02330.pdf)

**Questions:**

please comment on 2) above.

**Limitations:**

yes

---

> ### Author Rebuttal · Authors · 2023-08-07
>
> Thank you for taking the time to provide us with your valuable feedback, and also for recognizing our paper to be an “Excellent paper Excellent paper that brings together the concepts of consensus and equilibrium” with “Strong theory, interesting experiments.”
>
> **Comment**: "don't forget the conclusion in the final version of the paper."
>
> **Response**: We will be more than happy to add a conclusion section in our final version.
>
> **Comment**: “It is clear that in the long run, due to the strong law of large numbers, FP is such that agents within a population will form the same beliefs. Therefore, the point of view taken by the authors is to study a representative agent. I say: what could be interesting is the transient regime where agents beliefs have not yet converged. There, one could use the central limit theorem, and look at the interplay between, on the one side, consensus that has not been reached yet, and the convergence of populations to equilibria. You could also look at it with 2 different learning rates (one for intra, one for inter populations, I suggest you explore connections with this https://arxiv.org/pdf/2205.02330.pdf)”
>
> **Response**: This is indeed an excellent and deep direction for follow-up work that will probably require to bring in the mix a number of new technical ideas. Thank you very much for the suggestion! We will make sure to include in the discussion of interesting future work as well as add the relevant citation. Thank you again!

---

### Official Review · Reviewer_3rNt · 2023-07-05

**Soundness:** 2 fair
**Presentation:** 2 fair
**Contribution:** 2 fair
**Rating:** 4
**Confidence:** 2

**Summary:**

The authors defined a network population game, which is a multipartite network game where each partite is a population and agents in the same population do not interact with each other and only interact with agents from other populations. In this way, each population can be easily abstracted into a "super-agent", and each population has separate beliefs about different neighbor populations.

The authors show that when the agents perform interactions using the network population model and adopt a smooth fictitious play dynamic, the population's belief will gradually have lower variance and the mean belief will reach a quantal response equilibrium (QRE) in both weighted zero-sum games and exact potential network games.

**Strengths:**

1. The intention of this paper is good, which tries to address both the consensus and the convergence of multi-agent learning systems
2. The overall flow of this paper is easy to follow, and the authors conduct both theoretical and numerical studies

**Weaknesses:**

1. The justification for using the network population game is not sufficient. When this specific game model fits real-world problems, especially related to belief updates. For now, the purpose of this assumption seems like making the convergence and part much easier to study.
2. The reason for using the smooth fictitious play dynamics is also not sufficiently justified. The reason for including the \nu penalty term in the utility function should be provided (e.g., saying this is an entropy-based cost and why this is reasonable), and whether it is necessary for the convergence and consensus study should be elaborated.
3. No elaboration on how the \epsilon term and the A_ij influence the consensus and convergence
4. Lack of discussion on the limitations

**Questions:**

1. What is the intuition behind the perturbed payoff in Eqn (5)? Is this cost term a necessity for consensus and convergence?
2. Line 203 says "Agents maintain separate beliefs about different neighbor populations", where is this previously justified and why is this reasonable?


**Limitations:**

The authors did not discuss the limitations of this work in the paper.

I'm not sure if and how the population size of each population will influence the consensus and equilibrium outcome and whether there will be any fairness issues based on this. For now, I will not flag ethics review issues since I can't identify or exclude them.

It will be nice for the authors to add discussions on the limitations, at least in the appendix.

---

> ### Author Rebuttal · Authors · 2023-08-07
>
> Thank you for taking the time to provide us with your valuable feedback.
>
> **Comment**: “The justification for using the network population game is not sufficient…easier to study.”
>
> **Response**: Network population games model scenarios which are characterized by the presence of multiple interacting populations. A good survey is [1]. This survey has been cited nearly 3000 times indicating the wide applicability and interest in related questions. Arguably, one the simplest and very natural settings are interactions between two different type of agents (e.g. male and female) that gives rise to bi-matrix population games that correspond to single-edge networks in our case. We gave a brief description of some applications in Footnote 1 in the paper, however, we will be more than happy to expand on it. Overall, you can think of nodes as types and each agent needs to formulate beliefs about how other types of agents will act when they interact with them in the world.
>
> **Comment:** ”The reason for using the smooth fictitious play dynamics is also not sufficiently justified… elaborated.” and  “What is the intuition behind the perturbed payoff…convergence?”
>
> **Response**: We chose smooth fictitious play due to two reasons: (i) it is amongst the most well-known and commonly studied models in AI and game theory, and (ii) it is a belief-based learning model and naturally suits the investigation of consensus formation (consensus can be intuitively understood as the convergence of agents’ beliefs).
>
> The term $ v( x_i(k))$ in Equation 5 is standard for smooth fictitious play ([1] and the references therein); it represents the perturbation on the expected payoffs and can also be understood as the control cost to implementing a strategy. A typical form is an entropy-based one $v (x_i(k)) =  -\frac{1}{\beta}\sum_{s_i \in S_i} x_{is_i}(k)\ln (x_{is_i}(k))$. The rationale is that by adopting this form, an agent’s choice in response to his/her beliefs becomes probabilistic such that better responses are more likely than worse responses, but the best ones are not played with absolute certainty; this aligns with human’s bounded rationality and error-prone decision-making [2]. We will elaborate more on this in our revision.
>
> Importantly, a specific form of $ v( x_i(k))$ is NOT necessary for our study. We wrote in Line 192-193 that “all our results readily generalize to any function $v$ satisfying the above two standard assumptions.” We explained the two standard assumptions in Line 187-188.
>
> **Comment**: “No elaboration on how the \epsilon term and the A_ij influence the consensus and convergence.”
>
> **Response**: All our consensus and convergence results hold given a positive value of $\epsilon$. The payoff matrix $A_{ij}$ of the 2-player subgames has NO effect on consensus formation. We showed this formally in Theorem 1, and discuss this in Line 226 (where we wrote “Note that the above theorem makes no assumption about the 2-player subgames agents play”).
>
> Regarding the effect of the payoff matrices $A_{ij}$ on convergence to equilibrium, we defined in Equation 13 $A_{ij}$ that satisfies the weighted zero-sum property, and established the convergence in weighted zero-sum network (population) games in Theorem 2 and Theorem 4. Moreover, we defined in line 296 $A_{ij}$ that satisfies the exact potential property, and established the convergence in star-structure exact potential network (population) games in Theorem 3 and Theorem 5. In the Introduction, we summarized the effects of $A_{ij}$ on convergence to equilibrium in Line 109-116.
>
> **Comment**: “Lack of discussion on the limitations” and “The authors did not discuss the limitations … at least in the appendix.”
>
> **Response**:  Thank you! We will make sure to expand on this more intuitively, however, our theorems discuss precise conditions on the dynamics and games such that they apply. Expanding these results even further is an interesting direction for future work.
>
> In terms of the population size, we wrote in Line 77 and Line 158 that this paper considers “a population (continuum) of agents”. This assumption allows for better analytical tractability, and is standard in studies of population games [3] and mean-field games [4]. Empirically, we observed that if the number of agents is sufficiently large (e.g., more than hundreds) for each population, our theoretical findings still hold. We will discuss the above limitations in our revision.
>
> **Comment**: “Line 203 says…  why is this reasonable?”
>
> **Response**: We mentioned this in Line 174, where we wrote “Agent $k$ maintains a weight $\kappa^i_{js_j}(k)$ for each opponent strategy $s_j \in S_j$ of each population $j\in V_i$.” As agents maintain $\kappa$ for each population and form their beliefs based on $\kappa$ (Equation 3), they naturally maintain separate beliefs about different populations. We gave a real-world example for separate beliefs in Footnote 4, where we wrote “people form beliefs about the behaviours of taxi drivers vs non-professional drivers after observing the numerous driving behaviours on the road.”
>
> Thank you again for your constructive comments! We hope that you will consider revising the score if we have addressed your concerns satisfactorily.
>
> [1] J. Hofbauer, E. Hopkins. Learning in perturbed asymmetric games.Games Econ. Behav., 2005
>
> [2] R. McKelvey, T. Palfrey. Quantal response equilibria for normal form games.Games Econ. Behav., 1995
>
> [3] W. Sandholm. Population games and evolutionary dynamics. 2010
>
> [4] J. Lasry and P. Lions. Mean field games. Japanese journal of mathematics, 2007

---

> > ### Comment · Reviewer_3rNt · 2023-08-18
> > **More questions on the multi-partite network structure**
> >
> > Thank you for the rebuttal. I'm still not fully convinced that the multi-partite graph is a natural assumption, it reads like agents in the same population do not interact with each other. Could you please elaborate on whether your results can generalize to other network structures?

---

> > > ### Author Response · Authors · 2023-08-18
> > > **Reply to the questions on multi-partite network structure**
> > >
> > > Thank you for your question. We will answer in two ways. First, we will describe several examples of settings where agents of the same population do not interact with each other. Secondly, we will describe a reduction that allows us to model such intra-population interaction using our current model.
> > >
> > > There are numerous examples of multipopulation interaction where the related interaction is captured by a normal form game that is asymmetric and requires exactly one individual of each population to play the game. The prototypical such example is Men-Women interaction in a game like Battle of the Sexes (or variants thereof). Similarly, we can think of an ecosystem with a graph of predator/prey interactions. The fact that there is no self-interaction within a population would capture that there is no in-species cannibalism. Or a more human centric example would capture battle tactics in a multi-army combat setting. Combatants of the same army do not face each other. A digital analogue of this example would be an E-sport competition in a multi-player game where each let's say of 5 agents compete against each other in a winner take all match. Each agent is produced by a different company (i.e. DeepMind, OpenAI, etc) and the way these agents work e.g. in the Double Oracle PSRO[1] literature is that they are encode a distribution over different NNs agents each with different capabilities. So actually, each digital agent is best thought of as a large mixture of distinct agents.
> > >
> > > Now, we will point out how our current setting actually allows for interactions between agents of a single population. Take our current setting and for each node/population i create a copy i' that is connected to the corresponding set of agent copies as the original node i, with exactly the same set of games and the initial state of node i' is identical to that of node i. Now, create a symmetric two-player game between nodes i and i'. This will allow us to capture the intra-population interaction. By the symmetry of the setting, the initial symmetry between node i and node i's will be preserved for all time t>0 and this "mirror" population node allows us to capture such intra-population interaction within our current setting. We are happy to expand upon such ideas and in general it is known that such learning in network games can be adapted to allow for self-loops without significant changes in the underlying analysis (see e.g. [2]).
> > >
> > >
> > > [1] Lanctot et al. "A unified game-theoretic approach to multiagent reinforcement learning." Advances in neural information processing systems 30 (2017).
> > >
> > > [2] Boone et al. "From Darwin to Poincaré and von Neumann: Recurrence and cycles in evolutionary and algorithmic game theory." Web and Internet Economics: 15th International Conference, WINE 2019, New York, NY, USA, December 10–12, 2019, Proceedings 15. Springer International Publishing, 2019.

---

> > > > ### Author Response · Authors · 2023-08-19
> > > >
> > > > Thank you again for your questions and engaging us. Since we have addressed your remaining concern about allowing interaction within populations by detailing why our model is powerful enough to capture these effects, we hope that you would be willing to increase your score accordingly. Thank you again for the interesting questions!

---

### Official Review · Reviewer_2uya · 2023-07-06

**Soundness:** 3 good
**Presentation:** 3 good
**Contribution:** 3 good
**Rating:** 6
**Confidence:** 3

**Summary:**

This paper combines reaching consensus and convergence to equilibrium for network population games. Consider a network whose vertices correspond to a population. Edges between vertices (or populations) represent two-player sub-games between each pair of agents in these neighboring populations. The authors specifically focus on smoothed fictitious play for these two-player sub-games while agents seek to reach consensus in their beliefs about agents' policies in neighboring populations. In that sense, the approach is analogous to (or motivated by) the anonymous random matching interpretation of fictitious play dynamics to justify the myopic nature of the agents [Fudenberg and Kreps, Learning mixed equilibria. Games and Economic Behavior, 1993]. In particular, consider (large) populations of agents in each player role. Each period, all agents are matched to play the game and are told only to play in their own match. Agents are unlikely to play their current opponent again for a long time, even unlikely to play anyone who played anyone who played her. So, if the population size is large enough compared to the discount factor, it is not worth sacrificing current payoff to influence an opponent’s future play. In these populations, agents share their belief. The consensus dynamics presented serve this purpose. Therefore, the results are expected to hold even though I have not checked the proofs in detail.

**Strengths:**

- Convergence of SFP dynamics in weighted zero-sum network games and exact potential network games with star structure.

**Weaknesses:**

- There is no motivating example for the network population game formulation.
- Results for potential network games are presented only for star structure.

**Questions:**

- Can you provide a motivating example justifying the network population model in practice (specifically the 2-player sub-games)?
- What is the reason to restrict the network structure to star structure only in potential network games?

**Limitations:**

I have not identified any discussion about the limitations.

---

> ### Author Rebuttal · Authors · 2023-08-07
>
> Thank you for your time and for your constructive comments on our paper!
>
> **Comment**: “There is no motivating example for the network population game formulation” and “Can you provide a motivating example justifying the network population model in practice (specifically the 2-player sub-games)?”
>
> **Response**: Network population games are widely studied with numerous well-known variants. A good survey is [1]. This survey has been cited nearly 3000 times indicating the wide applicability and interest in related questions. Arguably, one the simplest and very natural settings are interactions between two different type of agents (e.g. male and female) that gives rise to bi-matrix population games that correspond to single-edge networks in our case. Moreover, the survey shows a lot of interest in focusing on interactions where each type of agents have only a handful of options (e.g. Battle of the Sexes, Prisoner's Dilemma etc).  We briefly described some applications in Footnote 1 in the paper. We sincerely appreciate your feedback, and we will gladly elaborate more on these and further examples in the introduction in our revision.
>
> **Comment**: “Results for potential network games are presented only for star structure” and “What is the reason to restrict the network structure to star structure only in potential network games?”
>
> **Response**:  Our proof of the Lyapunov function in the case of potential games depends on the cancellation of several terms, which currently utilises the assumption of a star structure. We agree that it is a very interesting direction to extend our results further. Nevertheless, we want to point out that several prior works in learning in games have utilised similar assumptions such as star network structure, e.g., [2-4].
>
>
> Thank you again for your positive assessment and constructive comments on our paper!
>
> [1] Gyorgy Szabo, Gabor Fath Evolutionary games on graphs, Physics reports, 2007.
>
> [2] Panageas, Ioannis, and Georgios Piliouras. "Average case performance of replicator dynamics in potential games via computing regions of attraction." Proceedings of the 2016 ACM Conference on Economics and Computation. 2016.
>
> [3] Sai Ganesh Nagarajan, et al. From chaos to order: Symmetry and conservation laws in game dynamics. In International Conference on Machine Learning, pages 7186–7196. PMLR, 2020.
>
> [4] Sela, Aner. "Fictitious play in ‘one-against-all’multi-player games." Economic Theory 14.3 (1999): 635-651.

---

> > ### Comment · Reviewer_2uya · 2023-08-18
> >
> > Thank you for the rebuttal. My concerns have not been addressed properly. My current understanding is that the two-population subgame structure is used for mathematical tractability. I am paraphrasing my questions for clarity:
> > - Clearly network population games are popular. I am asking for some tangible motivating examples for the 2-player subgame structure. More explicitly, can the authors provide some tangible examples motivating the reward function defined in Eq. (1) as a summation of two-population games? For example, related to the political opinion clusters from Footnote 1, what does Eq. (1) imply?
> > - Smooth fictitious play is known to converge equilibrium in exact potential games with finitely many players without any condition on the interconnections among them, e.g., see Section 4.2 in [Hofbauer and Sandholm, On the global convergence of stochastic fictitious play, Econometrica 2002]. If the population is acting identical to a single agent, what is the restriction preventing us to address network structures beyond star?
> >
> > I have an additional question: Do the results generalize to the cases where agents follow the classical fictitious play? Is there a particular reason to use the smoothed version apart from mathematical tractability?

---

> > > ### Author Response · Authors · 2023-08-18
> > > **Reply to the Comment by Reviewer 2uya**
> > >
> > > Thank you for your questions.
> > >
> > > **Reply to your first question**: We will describe two families of examples where 2-player/population subgame structure emerges.
> > > The first is actually an arbitrary congestion game with linear costs. Such settings, although maybe not obvious at a first glance are actually reducible to 2-player subgame structures. The two-agent interaction in entry $(i,j)$ captures the extra cost that e.g. each agent causes to the other one when the first player chooses path $i$ and the second player chooses path $j$. Since we have assumed that the costs increase linearly with the number of agents we can compute the total additive effect by merely summing up the costs over all such two agent interactions.
> > >
> > > Another example but now with adversarial incentives is that of tournament competition where every agent has to compete against every other agent and wants to maximize the number of heads-up matches they win. This is standard for example in chess. Now, if we want to have a population version of this game imagine an international chess tournament where every node/player is actually a nation and is represented by a team of players and players get matched randomly. Finally, if one wants to have a large population version of the above we can consider a similar version of the above chess competition but now between AI companies such as DeepMind, OpenAI etc each of which is submitting a single PSRO [1] type of mixture of NN agents.
> > >
> > > For the case of opinion formation imagine a cluster of nations that has to choose between two competing political philosophies/religions/coalitions etc but the safety of a nation depends on how many of its neighbors share the same attitudes.
> > >
> > > **Reply to your second question**:
> > > The model explored by Hofbauer and Sandholm is simpler than ours. Critically, the state space of our model includes both choice distributions $x$ and as well as beliefs $\mu$. In contrast, Hofbauer and Sandholm models only has choice distributions. Hence, arguments in this previous paper do not translate to ours and cannot say anything about the evolution of beliefs, which is a key aspect of our model. As we see in the proof of our convergence result, the Lyapunov functions (Equation 63 in the appendix) includes "mixed" terms that combine both $x$ and $\mu$ terms. Such complexities are not needed in the Hofbauer and Sandholm model.
> > >
> > > **Reply to your third question**:
> > > Studying different learning dynamics is a very interesting direction for future work. In this paper we focus on SFP and questions about other dynamics although interesting are out of the scope of the current work. Furthermore, we believe that in our setting FP would actually not be a good choice as in FP dynamics all agents are playing pure strategies, whereas our goal in this paper is to study the evolution of beliefs, which necessitates randomization at the level of the individuals.
> > >
> > > [1] Lanctot, Marc, et al. "A unified game-theoretic approach to multiagent reinforcement learning." Advances in neural information processing systems 30 (2017).

---

### Official Review · Reviewer_3EKR · 2023-07-10

**Soundness:** 2 fair
**Presentation:** 3 good
**Contribution:** 3 good
**Rating:** 6
**Confidence:** 3

**Summary:**

This paper examines the connection between the notions of consensus and equilibrium in a multi-agent system where multiple interacting sub-populations coexist, and it aims at answering below two central research questions in network (population) games scenario:
     [1] Are there natural multi-agent learning models that can achieve the best of both worlds—reaching consensus as well as convergence to equilibrium—in diverse settings?
     [2] How does the consensus formation process affect equilibrium selection in multi-agent learning?

The authors argue that consensus and equilibrium, the fundamental notions of these two fields, can be both understood as stability concepts in a multi-agent system where there co-exist multiple interacting sub-populations. In particular, consensus can be seen as an intricate component of intra-population stability, whereas equilibrium can be seen as encoding inter-population stability. The authors show that SFP (smooth fictitious play) algorithm can achieve consensus as well as convergence to equilibrium in a wide range of network population games (and unlike previous literature, here a coordinative reward structure is not a prerequisite for achieving
consensus). They also empirically shows that consensus formation process plays a crucial role in the seminal thorny problem of equilibrium's election in multi-agent learning (e.g., starts with the same initial mean belief, a large variance of initial beliefs results in a more desired equilibrium). Experiments were conducted for the scenario of Equilibrium Selection in Two-Population Stag Hunt Games.

**Strengths:**

1. The paper is organized and presented well and clearly, I found the manuscript is reader friendly.

2. This work extend existing literature in multiple directions and presented a couple of nontrivial novel theoretical results, which is quite beneficial to the research community. E.g., SFP in network (population) games has not been previously explored until this work, unifying consensus formation and learning in multi-agent games, etc., proving consensus without assuming a coordinative reward structure, etc..

3. There are quite some helpful/valuable elaborations/explanations about comparing this work with relevant literature and discussing the differences and advantages.

**Weaknesses:**

1. I'd like to see some discussions about the future research directions, and how this work could inspire/benefit other future research.

2. Regarding the figure 1 about impacts of variances of initial beliefs, it seems to be just plot of one example setting, and the empirical conclusion is not very convincing to me and I'd like to see some more clarification/elaboration/justification. For example, consider such a scenario where the starting mean belief is already the optimal value of a final desired steady state/equilibrium, it seems that if the staring variance is 0 (extreme case of small variance), then convergence to optimal results is already achieved, which is obviously better than a larger initial variance with the same starting mean belief, which is a counter example of the empirical conclusion of the paper saying that a larger initial variance (given same starting mean belief) is preferred. I would like to see either some rigorous mathematical proofs about such conclusion, or very comprehensive empirical studies on this before drawing the conclusions on this.

3. In page 4, it was mentioned that "This paper formally shows that the probability distribution over initial conditions can eventually degenerate to a point mass, and leveraging on this, presents a novel technique for proving the convergence of learning dynamics." It would be good to see some more elaborations on how this "novel technique" could be used for proving the convergence of learning dynamics in other relevant problems.

**Questions:**

Please see my questions/comments/suggestions in above section when talking about weakness.

---

> ### Author Rebuttal · Authors · 2023-08-07
>
> Thank you! We sincerely appreciate your diligent and thoughtful comments on our paper.
>
> **Comment**: “I'd like to see some discussions about the future research directions, and how this work could inspire/benefit other future research.”
>
> **Response**:  We appreciate your interest in future directions of our work! We will be more than happy to add a section at the end of the paper where we expand on them.
>
> **Comment**: “Regarding the figure 1 about impacts of variances of initial beliefs, it seems to be just plot of one example setting, and the empirical conclusion is not very convincing to me and I'd like to see some more clarification/elaboration/justification. ... before drawing the conclusions on this.”
>
> **Response**: Our main argument regarding our stag-hunt game experiments is that a larger initial variance can promote convergence to the payoff dominant equilibrium (S,S). However, this does not imply “a larger initial variance (given same starting mean belief) is preferred”.
>
> While Figure 1 presents two example belief distributions with the same initial mean belief, Figure 2 provides additional evidence covering *ALL* possible initial mean beliefs under the same game settings. In Figure 2, we demonstrated that increasing the variance of initial beliefs from 0 to 0.02, 0.05, and 0.1 expands the region of attraction of the payoff dominant equilibrium (S,S), allowing a wider range of initial mean beliefs to approach it. Based on the enlarged region of attraction, we concluded that a larger initial variance can promote convergence to the equilibrium (S,S). Importantly, this does not mean that a larger initial variance is always preferred. We will make sure to add further elaboration on our empirical results to make the above points more precise in our revision.
>
> **Comment**: “In page 4, … how this "novel technique" could be used for proving the convergence of learning dynamics in other relevant problems.”
>
> **Response**: By “novel technique”, we meant our approach for establishing the convergence result of smooth fictitious play in network population games. The novelty of our approach is largely characterized by two key steps: (i) we proved the variance of the belief distribution tends to zero in the limit, and (ii) we leveraged the zero-variance (i.e. consensus) property to extend the convergence result in classic network games to network population games. Our approach highlights the zero-variance/consensus property as a valuable tool for establishing the convergence of learning under population settings. For future research that studies learning dynamics under population settings, one can find inspiration in our approach by initially verifying the zero-variance/consensus property and subsequently leveraging this property to establish the convergence result. We will incorporate the above points into our revision.
>
> Thank you again for your positive feedback and constructive comments!

---

> > ### Comment · Reviewer_3EKR · 2023-08-18
> >
> > I've read the authors' rebuttal (which help provides some helpful clarifications) as well as all the reviews from other reviewers. I'd keep my rating unchanged as 6 with weak acceptance suggestion, taking into account all of them. I think the manuscript might be acceptable for publication here, but I won't push hard if other reviewer has strong objections on this.

---

> > > ### Author Response · Authors · 2023-08-19
> > >
> > > Thank you for your response and your continued support.

---

### Decision · Program_Chairs · 2023-09-21

**Decision:**

Accept (poster)

**Comment:**

The paper showed that smoothed fictitious play achieves both consensus and convergence to equilibrium in diverse population network games, which is an interesting result and well-presented. There was some concern regarding the generality of the setting (multi-population assumption) considered. I found the paper solid and cleared the bar of acceptance.